# RobIA: Robust Instance-aware Continual Test-time Adaptation for Deep Stereo

**Jueun Ko**[1*]     **Hyewon Park**[1*]     **Hyesong Choi**[2]     **Dongbo Min**[1†]
[1]Ewha Womans University    [2]Soongsil University
{jueun.ko, hwpark}@ewha.ac.kr  hyesong@ssu.ac.kr  dbmin@ewha.ac.kr
https://github.com/0ju-un/RobIA

## Abstract

Stereo Depth Estimation in real-world environments poses significant challenges due to dynamic domain shifts, sparse or unreliable supervision, and the high cost of acquiring dense ground-truth labels. While recent Test-Time Adaptation (TTA) methods offer promising solutions, most rely on static target domain assumptions and input-invariant adaptation strategies, limiting their effectiveness under continual shifts. In this paper, we propose **RobIA**, a novel *Robust, Instance-Aware* framework for Continual Test-Time Adaptation (CTTA) in stereo depth estimation. RobIA integrates two key components: (1) Attend-and-Excite Mixture-of-Experts (AttEx-MoE), a parameter-efficient module that dynamically routes input to frozen experts via lightweight self-attention mechanism tailored to epipolar geometry, and (2) Robust AdaptBN Teacher, a PEFT-based teacher model that provides dense pseudo-supervision by complementing sparse handcrafted labels. This strategy enables input-specific flexibility, broad supervision coverage, improving generalization under domain shift. Extensive experiments demonstrate that RobIA achieves superior adaptation performance across dynamic target domains while maintaining computational efficiency.

## 1   Introduction

Stereo Depth Estimation (SDE) is a fundamental task for 3D scene understanding, with applications in autonomous driving and robotics. While deep learning-based stereo approaches have achieved notable accuracy improvements [1, 2], their success relies largely on supervised training with dense ground-truth disparity maps, which are costly and labor-intensive to obtain. As a result, they are typically pre-trained on large-scale synthetic datasets [2] and later adapted to real world training datasets. However, they suffer from domain shifts by challenging conditions unseen in the training datasets, leading to performance degradation during inference. To deal with these challenges, Test-Time Adaptation (TTA) has recently emerged [3–6], aiming to adapt a model on-the-fly to unseen target domains during inference, generally conducted in an unsupervised manner.

Current TTA approaches for SDE [7, 8] operate under the assumption of a single, stationary target domain, overlooking more realistic scenarios where the domain distribution evolves over time, including changing weather, lighting, or scene structure. Continual Test-time Adaptation (CTTA) [9–12] has recently emerged as a framework for continuously adapting models to consistently evolving target domains, and has been applied to other vision tasks such as classification and semantic segmentation. In this regard, we explore a more practical and challenging CTTA setting for SDE. In general, CTTA introduces two major challenges: *catastrophic forgetting*, where the model gradually loses the knowledge acquired from source domains while adapting to new target domains, and *error*

---

*Equal contribution.
†Corresponding author.

39th Conference on Neural Information Processing Systems (NeurIPS 2025).

*accumulation*, where noisy pseudo-labels progressively degrade model performance. Addressing these issues requires a careful balancing between stability, to preserve prior knowledge, and plasticity, to accommodate new domain-specific variations.

To address the stability-plasticity trade-off, recent CTTA methods have explored Parameter-Efficient Fine-Tuning (PEFT) strategies [10, 12], which preserve the representation capacity of pre-trained backbones by freezing them, while enabling adaptation via a small set of trainable components such as prompt and meta-networks. These approaches have shown promising results in classification tasks, but their effectiveness diminishes in more complex settings such as semantic segmentation [13], which requires dense, spatially structured predictions. A key limitation lies in the input-invariant nature of standard PEFT modules where adapters or prompts are fixed for all inputs, making it difficult to capture instance-specific variations [14, 15]. This underscores the need for PEFT methods that dynamically adapt to each instance, particularly under continual domain shifts.

In addition to architectural adaptability, the quality of supervision plays a crucial role in effective adaptation. Existing methods for SDE commonly rely on photometric consistency loss [7, 8] or pseudo-labels generated by handcrafted stereo matching algorithms [16], which are considered relatively robust to domain shifts [17]. Prior works typically filter these pseudo-labels with confidence-based thresholding, since they are often unreliable in challenging regions such as occlusions, reflective surfaces, or low-texture areas. While this selective supervision improves pseudo-label quality, it introduces a critical drawback: the model is trained only on a subset of the input target domains, leading to over-reliance to confident pseudo labels and weak generalization to uncertain or structurally complex regions. These observations highlight the need for more comprehensive supervision signals that can cover the full input distribution and mitigate the risk of pseudo-label over-reliance during continual adaptation.

To enable instance-aware adaptation and improve pseudo-supervision under dynamic conditions, we propose Robust, Instance-Aware CTTA approach, termed **RobIA**, which is tailored for stereo depth estimation in continually shifting domains. RobIA addresses the limitations of conventional PEFT and proxy-labeling approaches through two key components. First, we introduce the Attend-and-Excite Mixture-of-Experts (AttEx-MoE), a compact yet effective MoE architecture that enables input-specific adaptation without updating the backbone. Inspired by selective channel excitation [18], AttEx-MoE dynamically activates the convolutional channel experts via a self-attention, conditioned on instance-aware global features. To reduce computational overhead, we constrain the attention operation to be row-wise along epipolar lines, leveraging stereo geometry while preserving long-range contextual reasoning. This design allows AttEx-MoE to maintain the efficiency of PEFT while introducing fine-grained, content-aware adaptability crucial for dense stereo predictions.

Second, we propose the Robust AdaptBN Teacher, a complementary PEFT-based model that enhances the coverage of pseudo-labels during adaptation. Prior work [3, 19] has presented an effective test-time adaptation mechanism by updating only the affine parameters of batch normalization layers, known as AdaptBN [20, 21]. Building on this, we leverage an AdaptBN-trained teacher model to complement the sparsity of handcrafted stereo pseudo-labels in low-confidence regions. Specifically, we adopt a hybrid supervision scheme: reliable pseudo-labels from handcrafted stereo matching algorithms are used in high-confidence areas, while predictions from the Robust AdaptBN Teacher supervise low-confidence regions. This dual-source guidance allows the model to retain the precision of proxy labels where they are reliable, while extending supervision coverage to previously ignored areas, thus promoting better generalization across the entire input space.

Our key contributions are summarized as follows: **(1)** We propose RobIA, a novel CTTA framework specifically designed for stereo depth estimation under dynamic domain shifts. **(2)** We introduce a parameter-efficient, instance-aware adaptation module (AttEx-MoE) that dynamically routes input through frozen convolutional experts using a lightweight row-wise self-attention mechanism. **(3)** We design a complementary PEFT-based (AdaptBN) teacher that provides pseudo-supervision in low-confidence regions, enhancing coverage and robustness of pseudo labels. **(4)** We propose a dual-source supervision scheme that combines reliable handcrafted stereo pseudo-labels with predictions from the AdaptBN Teacher, enhancing coverage and robustness of pseudo labels.

## 2 Related Works

**Test-Time Adaptation for Stereo Depth Estimation** Test-Time Adaptation (TTA) in stereo depth estimation (SDE) aims to adapt a model to new domains in an online or real-time manner without

access to source data or ground-truth labels. Early approaches include modularized model update [8], meta-learning-based adaptation [7], and pixel-wise focused adaptation [22]. Despite these advancements, prior stereo TTA approaches have mostly focused on *single-domain adaptation* or *long-term adaptation* using large sequences (typically each domain contains more than 2K frames) within a static domain. These approaches overlook practical scenarios in which domains evolve continuously over time. To address this gap, we introduce a continual test-time adaptation scenario for SDE that reflects temporally evolving domain distributions.

**Self-supervised Learning for Stereo Depth Estimation** A long-standing challenge in SDE is the low density and high acquisition cost of ground-truth labels. To overcome this, self-supervised learning has been widely adopted [23, 24] , commonly using the photometric loss between stereo pairs. However, this signal often suffers from matching ambiguities, such as occlusions and specular surfaces, leading to unreliable supervision. [24, 17] exploited traditional stereo algorithms to generate pseudo-labels, filtering the outliers with confidence measures. [25] further introduced a monocular completion network to distill hard-to-match regions from stereo matching. While effective, it requires a separate network and multiple inference steps, leading to significant computational overhead, making it impractical for online adaptation. In contrast, our approach leverages a robust teacher model with lower computational cost, offering reliable guidance even in challenging regions where handcrafted pseudo-labels are absent.

**Mixture-of-Experts** Mixture-of-Experts (MoE)[26, 27] has been widely used in various domains, including CTTA MoE enables dynamic selection of expert subnetworks via routing mechanism, making it effective for multi-task learning [28, 29] and continual learning [30, 31]. In CTTA, several studies have explored parameter-efficient fine-tuning (PEFT) approaches that integrate MoE modules into pre-trained backbones [32, 33], enabling domain adaptation with minimal trainable parameters. However, most of this research has been conducted on Transformer-based architectures, and the application of MoE within CNNs remains relatively unexplored. DeepMoE [34] is a common approach in CNN-based architecture to treat individual channels as experts, allowing fine-grained modulation of feature representations and improving model sparsity. Our work is motivated by this underexplored direction, proposing a CNN-compatible MoE design that supports instance-aware adaptation under continual domain shifts.

## 3 Preliminaries

**Continual Test-time Adaptation** In CTTA paradigm, we are given a model pre-trained on a source dataset $(\mathcal{X}_S, \mathcal{Y}_S)$. The goal is to adapt the model to multiple unlabeled target data distribution $\mathcal{X}_T = \{\mathcal{X}_{T_1}, \mathcal{X}_{T_2}, \ldots, \mathcal{X}_{T_n}\}$ during deployment, where $n$ represents the number of unseen domains. When the target domain data $\mathcal{X}_{T_i}$ consists of $N_i$ target samples for $i = 1, \ldots, n$, for simplicity, we denote $(I_t^L, I_t^R)$ as the $t^{\text{th}}$ target stereo image pair for $t = 0, \ldots, |\mathcal{X}_T| - 1$, where $|\mathcal{X}_T| = \Sigma_{i=1}^n N_i$. Similarly, in the following sections of this paper, we omit the domain notation $\mathcal{X}_{T_i}$ as all target domains are considered to be integrated into a single sequence. Therefore, at each time step $t$, the stereo model predicts disparity map from a single stereo image pair $(I_t^L, I_t^R)$, and updates its parameters before proceeding to the next input.

**Pseudo-supervision for SDE** Since ground-truth labels are unavailable at test time, the model must be trained in a *self-supervised manner*. Following prior work [17], we obtain the handcrafted disparity map $D_{\text{proxy}}$ as a pseudo-label using the traditional stereo matching algorithm, Semi-Global Matching (SGM) [16]. Then, $D_{\text{proxy}}$ is filtered by the confidence threshold $\varepsilon$, where confidence $c_t(p)$ is calculated via left-right consistency at the corresponding pixel $p$. Consequently, the mask $\mathcal{M}_{\text{valid}}$, which indicates the reliable region of $D_{\text{proxy}}$ (*i.e.* valid region), is denoted as follows:

$$\mathcal{M}_{\text{valid}}(p) = \begin{cases} 1 & \text{if } c_t(p) \geq \varepsilon \\ 0 & \text{otherwise} \end{cases} \tag{1}$$

$\mathcal{M}_{\text{invalid}} = 1 - \mathcal{M}_{\text{valid}}$, indicating the unreliable region of $D_{\text{proxy}}$ (*i.e.* invalid region).

**Mixture-of-Experts in CNN** Prior work [34] applying Mixture-of-Experts (MoE) to convolutional neural networks (CNNs) formulates $C$ convolutional kernels (*i.e.* the output channels of the previous layer) as an individual expert $E_i$ for $i = 1, .., C$, and introduces a gating network $G$ to compute a

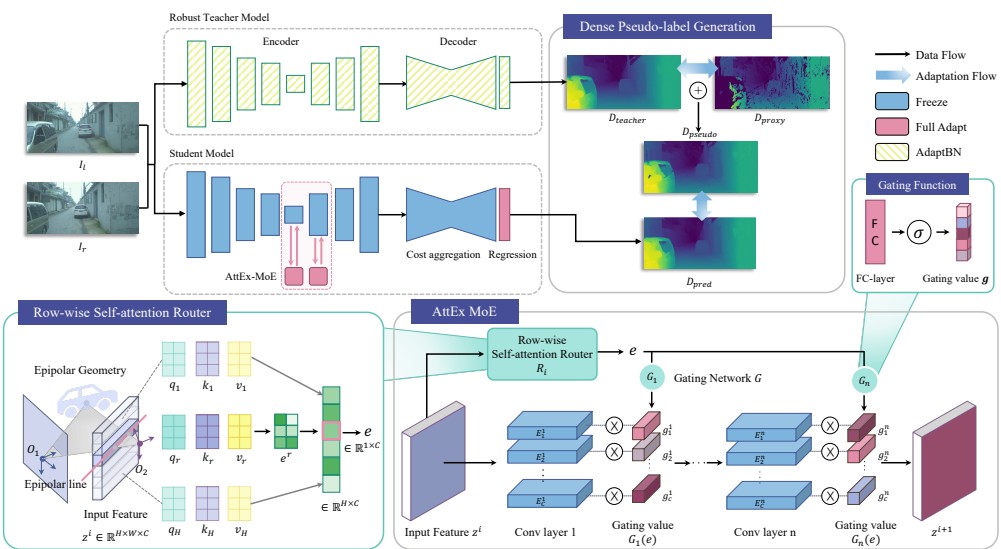

Figure 1: **The Overview of RobIA.** During test time, the student model is trained using dense pseudo-labels generated by combining sparse handcrafted proxy $D_{\text{proxy}}$ with Robust Teacher prediction $D_{\text{teacher}}$, ensuring stable adaptation under dynamic conditions. AttEx-MoE integrates a row-wise self-attention router and gating network $G$ into deep encoder blocks. The row-wise self-attention router extracts global context from an input feature map $z$, which is subsequently processed by a gating network. The student backbone is kept frozen, and only the router, gating network, and the regression parameters of the decoder are updated.

weighted combination of expert outputs. The output feature $y$ is defined as:

$$y = \sum_{i=1}^{C} g_i \cdot E_i(x), \tag{2}$$

where $\boldsymbol{g} \in \mathbb{R}^C$ is the gating values obtained from the gating network $G$, with $g_i$ being the weight assigned to the $i^{\text{th}}$ expert $E_i(x)$. In this case, the gating network $G$ is implemented as a multi-headed sparse gating network that takes a shallow embedding $e$ of the *raw input image* and produces a channel-wise activation score. $G(e)$ is obtained by projecting the embedding $e$ into a score vector via a learnable weight matrix $W_g$, followed by ReLU activation:

$$G(e) = \text{ReLU}(W_g \cdot e). \tag{3}$$

This formulation enables input-dependent dynamic selection of feature channels, improving both model sparsity and representational flexibility.

## 4 Proposed Method

### 4.1 Motivation and Overview

CTTA presents several key challenges, notably catastrophic forgetting and error accumulation. Parameter-efficient fine-tuning (PEFT) methods address these issues by freezing pretrained weights, preserving source knowledge. However, in conventional PEFT-based CTTA approaches, an efficient module that adapts to the target domain is added to the original feature, performing the uniform transformation on all inputs. This rigid transformation is insufficient for handling various input variations, particularly in dense predictions under domain shifts. Moreover, pseudo-labels from handcrafted stereo algorithms are often consistently filtered in the unreliable supervision regions due to inherent matching difficulties. Despite their domain-agnostic properties, they lead to over-reliance on reliable regions, misdirected adaptation elsewhere, and ultimately degrading overall performance.

To overcome these limitations, we propose a novel CTTA framework, **RobIA** (see Fig. 1). RobIA is designed to preserve the rich representational capacity of source knowledge while enabling stable and effective adaptation through the **Attend-and-Excite Mixture-of-Experts (AttEx-MoE)** architecture. This module extracts instance-specific features via a row-wise self-attention router and dynamically

excites frozen channel experts, allowing the model to adapt flexibly without modifying the pretrained backbone. To further address the challenge of biased and sparse pseudo-labels, we additionally introduce the **Robust AdaptBN Teacher**. It provides a hybrid supervision strategy that combines reliable handcrafted pseudo-labels with predictions from the robust teacher model.

## 4.2 Input-Aware Mixture-of-Experts via Attend-and-Excite

Mixture-of-Experts (MoE) in CNNs consists of a shallow embedding network and gating network with ReLU activation to treat each convolutional kernel as an individual expert and introduce sparsity into the model. However, this approach presents two key limitations: (1) shallow convolutional layers used in the embedding network lack sufficient spatial context for optimal expert selection, and (2) ReLU-induced sparsity limits representational capacity, especially when the backbone is frozen. To address these challenges, we propose **Attend-and-Excite Mixture-of-Experts (AttEx-MoE)**, a novel MoE design tailored for PEFT-based adaptation in stereo depth estimation. AttEx-MoE enables instance-aware, channel-wise modulation under continual domain shifts.

**Row-wise Self-Attention Router** A key component of AttEx-MoE is the row-wise self-attention router, which extracts global context to guide instance-aware expert excitation. Since convolutions are effective at capturing local patterns but struggle with modeling long-range dependencies, prior work [18] identified this limitation and proposed global average pooling (GAP) to create channel-wise features. However, GAP has the drawbacks of treating all spatial positions uniformly and ignoring spatial importance. Therefore, we employ a self-attention mechanism in the gating network to explicitly capture inter-channel dependencies and their spatial relationships. This allows Attend-and-Excite operations to adaptively excite the most relevant features for each target instance. Moreover, we apply 1D self-attention along each epipolar line (*i.e.*, row) of the feature map by leveraging stereo geometry [35], significantly lowering computational overhead while maintaining global context.

As shown in Fig. 1, the encoder of our model consists of $N$ blocks, with the AttEx-MoE module applied to the final, deepest encoder block (*i.e.*, at $1/32$ resolutions) and the corresponding upsampling module connected via skip connections. In each block, the row-wise self-attention router $R_i$ for $i = 1, ..., N$ computes the gating input $e$ from the input feature map $z^{i-1}$, which is then passed to the gating network $G$ in each convolutional layer.

$$e = \frac{1}{H} \sum_{r=1}^{H} e_r, \tag{4}$$

where $e_r = \text{softmax}\left(\frac{q_r k_r^\top}{\sqrt{d}}\right) v_r$ is computed for $r \in [1, H]$ using $q_r = z_r W_q$, $k_r = z_r W_k$, and $v_r = z_r W_v$. Here, $z_r \in \mathbb{R}^{W \times C}$ denotes the $r$-th row of the feature map $z \in \mathbb{R}^{H \times W \times C}$, and the attention is computed independently for each row. The attention score $e_r$ obtained along the epipolar line is averaged over the height dimension $H$ to compute the final gating input $e$.

**Expert Excitation via Sigmoid** In [34], each expert output $E_i(x)$ is scaled by a gate value $G(x)$ produced via ReLU activations, which suppresses certain experts by zeroing their outputs, as explained in (2) and (3). While these sparse expert outputs can enhance generalization [36], this sparsity may reduce expressivity when the backbone is frozen in the PEFT setting.

Therefore, we adopt a Sigmoid-based soft gating mechanism, in contrast to the commonly used ReLU. This design allows all experts to be activated to varying degrees rather than enforcing hard selection of a subset of experts, promoting richer expert combinations and diverse representations conditioned on each input instance. We find that soft gating is particularly effective in the PEFT setting (See Tab. 5), as it maximizes the utilization of all available experts with limited learning capacity. The gating mechanism is defined with the Sigmoid activation $\sigma$ as:

$$G(e) = \sigma(W_g \cdot e). \tag{5}$$

## 4.3 Dense Pseudo-label Generation via Dual Supervision

For effective adaptation to the target domain, the design of the supervision plays a critical role. Pseudo-labels generated by traditional stereo algorithms are typically sparsified through reliability-based thresholding [17], which leads to over-reliance on sparse supervision and consequently limits performance gains in CTTA. To address this, we propose a dense pseudo-label generation strategy that

combines the domain-agnostic reliability of handcrafted pseudo-labels with the learning capability of a learnable teacher model.

AdaptBN [3, 20, 21] adapts only the affine parameters (scale $\gamma$, shift $\beta$) of batch normalization layers, enabling low-dimensional, channel-wise feature modulation and stable adaptation. We leverage the AdaptBN-based teacher model to complement unreliable regions in sparse, handcrafted pseudo-labels, producing dense pseudo-labels as supervision for student model. This mitigates the student's over-reliance on sparse pseudo labels and reduces performance degradation in unreliable regions.

**Why AdaptBN?** Most CTTA frameworks [9, 11] avoid stochastically-updated teacher model, as it is expected to provide stable supervision with minimal computational and memory overhead. Accordingly, common CTTA designs adopt Mean Teacher models updated via exponential moving average (EMA) [37] or fixed, source-trained models [38]. However, stereo depth estimation task presents a unique setting where handcrafted stereo algorithms can yield reliable pseudo-labels (*i.e.* proxy), reducing the need for stability-focused teachers. In this context, the teacher must not only stabilize but also generalize beyond the limitations of the sparse proxy, particularly in regions where handcrafted labels are unreliable.

Since AdaptBN performs adaptation via low-dimensional affine transformations, it offers a controlled yet expressive adaptation mechanism for test-time adaptation. This makes AdaptBN spatially robust, allowing the teacher to effectively adapt to the target domain while reducing the model's over-reliance to the sparsely provided proxy labels. In contrast, EMA-based teachers are less suitable in this context. Although they maintain stability by updating weights gradually, they are prone to error accumulation when the student produces biased predictions. In such cases, EMA teachers tend to reinforce incorrect predictions, failing to provide correct guidance to the student [39]. We further validate these claims through analysis and ablation studies, which demonstrate AdaptBN teacher's contributions to both stability and adaptive refinement, especially in regions where handcrafted labels are unreliable.

### 4.4 Continual Test-time Adaptation Process

**Model Initialization** We insert the lightweight AttEx-MoE module into a source-trained base model. Following recent CTTA studies [32], we train the AttEx-MoE module on the labeled source dataset while keeping the backbone frozen. This short *warm-up* phase allows the MoE module to learn instance-aware routing behavior without altering the core representations, providing a stable initialization for CTTA. During the supervised warm-up phase, we trained the model in the same way as DeepMoE [34]. For test-time adaptation, we freeze the base network and only update the AttEx-MoE module and the regression parameters of the decoder.

**Total Loss** The overall loss function is as follows:

$$\mathcal{L} = \mathcal{L}_{\text{proxy}} + \lambda \mathcal{L}_{\text{teacher}} \tag{6}$$

$$\mathcal{L}_{\text{proxy}} = \mathcal{M}_{\text{valid}} \cdot smooth_{L1}(D_{\text{proxy}} - D_{\text{pred}}), \tag{7}$$

$$\mathcal{L}_{\text{teacher}} = \mathcal{M}_{\text{invalid}} \cdot smooth_{L1}(D_{\text{teacher}} - D_{\text{pred}}), \tag{8}$$

where $D_{\text{proxy}}$ and $D_{\text{teacher}}$ denote the pseudo-labels generated by the handcrafted stereo algorithm [16] and the AdaptBN teacher model, respectively. The predicted disparity map $D_{\text{pred}}$ is supervised by two loss terms, $\mathcal{L}_{\text{proxy}}$ and $\mathcal{L}_{\text{teacher}}$, corresponding to each pseudo-label source. Incorporating supervision from $D_{\text{teacher}}$ helps regularize the student model, mitigating over-reliance on sparse handcrafted labels and preventing performance drops in regions lacking reliable pseudo-labels. $\lambda$ is a loss weight, which controls the influence of teacher predictions in our dense pseudo-label formulation.

## 5 Experiments

**Datasets.** To simulate TTA and CTTA scenarios, following prior work [8], all experiments were conducted on well-renowned stereo benchmarks, including KITTI RAW [40], DrivingStereo [41], and DSEC [42]. These datasets cover various conditions, such as different weather scenarios and urban cityscapes in both daylight and nighttime. The synthetic Flyingthings3D, part of the synthetic SceneFlow dataset [2] was used to pretrain the stereo model before test time. Sparse pseudo-labels $D_{\text{proxy}}$ were obtained via Semi-Global Matching (SGM) [16], followed by a left-right consistency check. The effective label density varies across datasets—roughly 92% for KITTI RAW, 72% for DrivingStereo, and 45% for DSEC, highlighting the need for robust dense supervision, especially in relatively sparser datasets such as DrivingStereo and DSEC.

Table 1: Performance comparison of Continual Test-time Adaptation on **DrivingStereo** benchmark over 10 rounds. To save space, only 1st and 10th round scores are written. **Bold** denotes best and *AT* denotes our method with dense pseudo-label $D_{\text{teacher}}$.

| Method | Round Condition Adapt. | 1 dusky D1-all | EPE | cloudy D1-all | EPE | rainy D1-all | EPE | 10 dusky D1-all | EPE | cloudy D1-all | EPE | rainy D1-all | EPE | All ↓ Mean D1-all | EPE |
|---|---|---|---|---|---|---|---|---|---|---|---|---|---|---|---|
| MADNet 2 [45] | (a) no adapt. | 13.24 | 1.69 | 6.56 | 1.22 | 11.51 | 2.18 | 13.24 | 1.69 | 6.56 | 1.22 | 11.51 | 2.18 | 10.44 | 1.70 |
| | (b) FT | 5.08 | 1.06 | 4.82 | 1.09 | 6.4 | 1.43 | 6.04 | 1.66 | 5.85 | 2.06 | 7.13 | 1.9 | 6.13 | 1.73 |
| | (c) MAD++ | 6.46 | 1.15 | 4.36 | 1.04 | 6.01 | 1.29 | 5.79 | 1.60 | 6.04 | 2.01 | 7.28 | 1.89 | 5.86 | 1.41 |
| CoEx [43] | (d) no adapt. | 5.53 | 1.14 | 3.55 | 0.99 | 7.61 | 1.64 | 5.53 | 1.14 | 3.55 | 0.99 | 7.61 | 1.64 | 5.56 | 1.26 |
| | (e) AdaptBN | 5.16 | 1.11 | 3.13 | 0.93 | 6.14 | 1.41 | 2.71 | 0.85 | 2.36 | 0.8 | 3.32 | 1.03 | 3.08 | 0.94 |
| | (f) FT | 5.25 | 1.11 | 2.98 | 0.91 | 5.81 | 1.37 | 3.05 | 0.88 | 2.48 | 0.81 | 3.63 | 1.09 | 3.04 | 0.92 |
| | (g) FT + AT | 5.09 | 1.1 | 3.01 | 0.91 | 5.83 | 1.38 | 2.63 | **0.84** | 2.33 | **0.79** | 3.08 | **0.99** | 2.93 | 0.91 |
| EcoTTA [12] | (h) MetaNet | 4.61 | 1.05 | 2.89 | 0.89 | **4.21** | 1.17 | 3.42 | 0.93 | 2.69 | 0.87 | 4.46 | 1.26 | 3.07 | 0.95 |
| **RobIA (ours)** | (i) AttEx-MoE | **4.01** | **1.01** | 2.4 | **0.84** | 4.44 | **1.13** | 2.72 | 0.88 | 2.29 | 0.84 | 3.89 | 1.22 | 2.98 | 0.97 |
| | (j) AttEx-MoE + AT | 4.28 | 1.03 | 2.4 | 0.84 | 4.54 | 1.16 | **2.4** | **0.84** | **2.24** | 0.82 | **3.02** | 1.00 | **2.77** | **0.91** |

For CTTA, we constructed a new benchmark by sampling 500 frames per domain from existing TTA sequences, constructing a short sequence with frequent domain shifts. Each cycle consists of 3–4 domains and is repeated over 10 rounds to simulate long-term adaptation with recurring conditions. Specifically, we obtained a sequence of dusky→cloudy→rainy for DrivingStereo, night1 to night4 for DSEC, and city→residential→campus→road for KITTI RAW. Unlike prior works that simulate long-term shifts over 44K frames [17], our CTTA setting imposes more rapid domain adaptation within short sequences, reflecting real-world constraints where environmental changes occur frequently and previously seen conditions may reappear. We include TTA results for all datasets, additional results on CTTA, and ablation studies on pseudo-supervision in the supplementary material.

**Implementation Details.** We use CoEx [43], a compact and real-time stereo network using MobileNetV2 [44] backbone, as our base architecture. Following prior work [45], we retrained the model on the synthetic source datasets with strong data augmentations to improve generalization. All experiments were conducted on NVIDIA A6000 and RTX 3090 GPUs and further implementation details, including hyper parameters, are in the supplementary material.

*no adapt.* denotes the source-trained model without adaptation. We additionally evaluated two variants of MADNet2 [45]: *FT*, which updates all model parameters, and *MAD++*, which applies modular updates. All PEFT-based CTTA methods, including *AdaptBN*, *MetaNet*, and our proposed *AttEx-MoE*, were implemented on top of the CoEx for a fair comparison. AdaptBN tunes the affine parameters of batch normalization layers, MetaNet tunes the meta network from EcoTTA [12], and AttEx-MoE updates the lightweight gating module for expert selection. In all cases, the decoder's regression parameters were jointly updated. *AT* setting uses dense pseudo-labels generated from our AdaptBN teacher.

**Evaluation Metrics.** We reported End-Point Error (EPE) and D1-all that measures the percentage of pixels with absolute disparity error exceeding 3 pixels and 5% of the ground truth. We adopt the standard online adaptation protocol: the model predicts each frame before updating, then uses that frame to adapt before moving to the next, reflecting deployment without access to ground truth.

## 5.1 Main Results

**CTTA Experiments.** Tab. 1 presents the 10-rounds CTTA performance on the DrivingStereo. Our method (h) consistently outperforms all baselines across different weather conditions and adaptation rounds on DrivingStereo. Compared to MADNet-based experiments (a)-(c), which is a state-of-the-art stereo TTA method, our approach yields substantial improvements in all domains (dusky, cloudy, rainy), demonstrating stronger generalization and adaptability under continual shifts.

While full tuning approach (f) achieves performance gains in the early rounds of the experiment, it suffers from performance degradation due to error accumulation and forgetting over time. Parameter-efficient tuning methods including (e) and (h) tend to preserve source knowledge more effectively, leading to more stable performance across rounds. However, these approaches often exhibit limited adaptation performance due to the restricted capacity of parameter-efficient tuning. For instance, (h) EcoTTA–which adapts meta network–shows reduced effectiveness on structured prediction tasks such

Table 2: Performance comparison of Continual Test-time Adaptation on **DSEC** benchmark over 10 rounds. To save space, only 1st and 10th round scores are written. **Bold** denotes best and *AT* denotes our method with dense pseudo-label $D_{\text{teacher}}$.

| Method | Condition Adapt. | Night#1 D1-all | EPE | Night#2 D1-all | EPE | Night#3 D1-all | EPE | Night#4 D1-all | EPE | Night#1 D1-all | EPE | Night#2 D1-all | EPE | Night#3 D1-all | EPE | Night#4 D1-all | EPE | Mean D1-all | EPE |
|---|---|---|---|---|---|---|---|---|---|---|---|---|---|---|---|---|---|---|---|
| | | **1** | | | | | | | | **10** | | | | | | | | **All↓** | |
| MADNet 2 | (a) no adapt. | 8.38 | 1.80 | 14.71 | 2.37 | 11.00 | 1.86 | 11.82 | 1.85 | 8.38 | 1.80 | 14.71 | 2.37 | 11.00 | 1.86 | 11.82 | 1.85 | 11.48 | 1.97 |
| | (b) FT | 4.7 | 1.24 | 7.79 | 1.49 | 5.97 | 1.31 | 6.33 | 1.31 | 3.57 | 1.11 | 7.34 | 1.4 | 5.62 | 1.25 | 5.8 | 1.24 | 5.71 | 1.27 |
| | (c) MAD++ | 5.52 | 1.34 | 8.43 | 1.53 | 6.21 | 1.34 | 6.66 | 1.33 | 3.89 | 1.16 | 7.53 | 1.42 | 5.7 | 1.27 | 5.89 | 1.27 | 6.04 | 1.32 |
| CoEx | (d) no adapt. | 6.10 | 1.38 | 12.24 | 1.94 | 8.34 | 1.58 | 8.05 | 1.50 | 6.10 | 1.38 | 12.24 | 1.94 | 8.34 | 1.58 | 8.05 | 1.50 | 8.68 | 1.60 |
| | (e) AdaptBN | 4.96 | 1.22 | 8.47 | 1.49 | 4.59 | 1.11 | **4.67** | **1.09** | 2.97 | 1.04 | 6.07 | 1.27 | 4.32 | 1.1 | 4.45 | 1.11 | 4.54 | 1.13 |
| | (f) FT | 4.99 | 1.23 | 8.41 | 1.48 | 4.66 | 1.12 | **4.67** | 1.1 | 3.00 | 1.04 | 6.24 | 1.28 | 4.44 | 1.11 | 4.57 | 1.12 | 4.59 | 1.13 |
| | (g) FT + AT | 5.11 | 1.24 | 8.76 | 1.52 | 4.73 | 1.13 | 4.73 | 1.1 | **2.87** | **1.01** | 5.86 | 1.24 | **4.02** | 1.06 | 4.15 | 1.07 | **4.38** | **1.10** |
| EcoTTA | (h) MetaNet | 4.36 | 1.17 | **7.17** | **1.39** | 4.69 | 1.13 | 5.33 | 1.18 | 3.46 | 1.08 | 6.71 | 1.36 | 4.71 | 1.14 | 5.28 | 1.19 | 5.13 | 1.21 |
| **RobIA (ours)** | (i) AttEx-MoE | **4.33** | 1.17 | 7.92 | 1.46 | **4.38** | **1.1** | 4.79 | 1.12 | 3.00 | 1.03 | 5.69 | 1.23 | 4.23 | 1.09 | 4.61 | 1.11 | 4.47 | 1.12 |
| | (j) AttEx-MoE + AT | 4.45 | **1.17** | 8.18 | 1.48 | 4.75 | 1.15 | 4.97 | 1.12 | 2.96 | **1.01** | **5.65** | **1.22** | 4.11 | **1.05** | 4.34 | 1.08 | 4.46 | 1.11 |

as stereo depth estimation. In contrast, our approach (i) AttEx-MoE enables input-dependent expert routing and selective feature excitation, achieving both robust and adaptive test-time performance, leading to improved adaptation accuracy.

Incorporating dense pseudo-labels via the AdaptBN teacher (denoted *AT*) further enhances long-term stability. As shown in 10$^{\text{th}}$-round results, models trained with dense supervision ((g) and (j)) exhibit notably lower error rates after long-term adaptation compared to their sparse-only counterparts. This underscores the importance of broader supervision coverage, as our dual-source dense pseudo-label strategy helps prevent the model from focusing exclusively on the reliable regions of sparse handcrafted labels.

Tab. 2 reports results on the DSEC benchmark, which includes challenging nighttime conditions. Our method again outperforms MADNet2 and other baselines, maintaining strong adaptation performance. Notably, both FT and MoE variants benefit from AdaptBN-based supervision, showing improved accuracy and stability by round 10. For instance, in Night#4, D1-all error drops from 4.57 to 4.15 in (f) and (g), and from 4.61 to 4.34 in (i) and (j), respectively. Moreover, these results highlight that AttEx-MoE tuning provides greater adaptability than other PEFT-based methods ((e) and (h)), while achieving comparable performance to full model tuning with improved efficiency.

Table 3: Performance comparison of Continual Test-time Adaptation on **KITTI RAW** benchmark over 10 rounds. To save space, only 1st and 10th round scores are written. **Bold** denotes best and *AT* denotes our method with dense pseudo-label $D_{teacher}$.

| Method | Condition Adapt. | City D1-all | EPE | Residential D1-all | EPE | Campus$^{(\times 2)}$ D1-all | EPE | Road D1-all | EPE | City D1-all | EPE | Residential D1-all | EPE | Campus$^{(\times 2)}$ D1-all | EPE | Road D1-all | EPE | Mean D1-all | EPE |
|---|---|---|---|---|---|---|---|---|---|---|---|---|---|---|---|---|---|---|---|
| | | **1** | | | | | | | | **10** | | | | | | | | **All↓** | |
| MADNet 2 | (a) no adapt. | 3.83 | 1.08 | 3.06 | 1.09 | 5.43 | 1.12 | 3.34 | 1.05 | 3.83 | 1.08 | 3.06 | 1.09 | 5.43 | 1.12 | 3.34 | 1.05 | 3.92 | 1.09 |
| | (b) FT | 1.21 | 0.92 | 0.96 | 0.93 | 2.06 | 0.82 | 1.12 | 0.86 | 0.91 | 0.88 | 0.79 | 0.91 | 1.56 | 0.75 | 1.01 | 0.84 | 1.13 | 0.85 |
| | (c) MAD++ | 1.51 | 0.98 | 0.95 | 0.94 | 2.28 | 0.87 | 1.22 | 0.88 | 0.92 | 0.87 | 0.85 | 0.92 | 1.72 | 0.77 | 1.08 | 0.88 | 1.24 | 0.88 |
| CoEx | (d) no adapt. | 2.08 | 0.99 | 1.75 | 0.96 | 2.89 | 0.96 | 2.79 | 1.00 | 2.08 | 0.99 | 1.75 | 0.96 | 2.89 | 0.96 | 2.79 | 1.00 | 2.38 | 0.98 |
| | (e) AdaptBN | 1.09 | **0.86** | 0.82 | **0.9** | 1.5 | 0.74 | 1.05 | **0.85** | 0.66 | **0.84** | 0.72 | **0.89** | 1.25 | **0.7** | 0.86 | 0.82 | 0.91 | **0.82** |
| | (f) FT | 1.04 | **0.86** | 0.82 | 0.92 | 1.33 | 0.73 | **0.97** | 0.86 | 0.63 | **0.84** | 0.68 | 0.90 | 1.16 | 0.71 | 0.81 | **0.81** | 0.86 | 0.82 |
| | (g) FT + AT | 1.06 | 0.87 | **0.79** | 0.91 | **1.27** | **0.72** | 1.03 | **0.85** | 0.55 | **0.84** | **0.6** | **0.89** | 1.13 | **0.7** | **0.8** | 0.82 | **0.82** | **0.82** |
| EcoTTA | (h) MetaNet | **0.94** | 0.87 | 0.96 | 0.92 | 1.65 | 0.77 | 1.29 | 0.87 | 1.24 | 0.92 | 1.17 | 0.95 | 1.78 | 0.83 | 1.69 | 0.91 | 1.33 | 0.88 |
| **RobIA (ours)** | (i) AttEx-MoE | 1.18 | 0.88 | 0.97 | 0.92 | 1.45 | 0.75 | 1.26 | 0.86 | 0.76 | 0.85 | 0.8 | **0.89** | 1.32 | 0.74 | 1.07 | 0.83 | 1.06 | 0.84 |
| | (j) AttEx-MoE + AT | 1.21 | 0.89 | 1.04 | 0.92 | 1.51 | 0.75 | 1.41 | 0.86 | 0.78 | 0.86 | 0.77 | **0.89** | 1.36 | 0.75 | 1.19 | 0.84 | 1.09 | 0.84 |

Tab. 3 reports the CTTA performance on the KITTI RAW. Our method outperforms MADNet2, a state-of-the-art stereo TTA method, maintaining strong adaptation performance throughout the experiment. In the case of KITTI RAW, more than 90% of the handcrafted pseudo-labels are considered reliable, as the dataset primarily consists of daytime urban scenes that are less challenging for stereo matching compared to weather-affected or nighttime scenes. This leads to relatively stable adaptation compared to other benchmarks. However, (g) FT + AT benefit from the additional dense

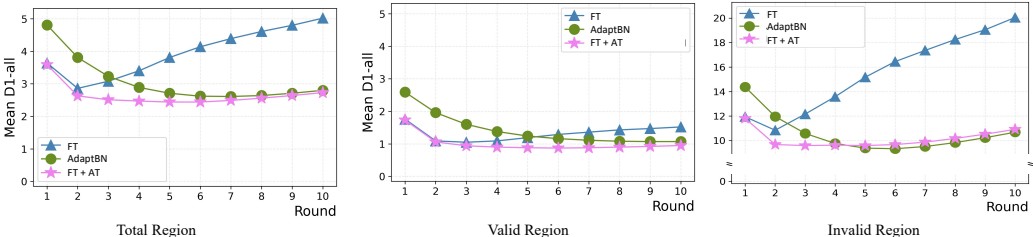

Figure 2: **D1-all error rate in different pseudo-label regions.** D1-all error rate over 10 adaptation rounds in different pseudo-label regions. We separate evaluation into (left) the entire image, (middle) regions with valid handcrafted pseudo-labels, and (right) regions without reliable supervision (invalid).

pseudo-labels, which improve model adaptability by further expanding the supervision coverage, resulting in better performance by round 10.

## 5.2 Analyses

**Performance degradation due to Sparse Pseudo-labels.** To understand the limitations of sparse handcrafted pseudo-labels, we evaluate model performance under continually shifting weather conditions by separately analyzing regions with reliable (valid) and unreliable (invalid) pseudo-labels with higher learning rate($2e-6$). As shown in Fig. 2, models trained solely with sparse supervision show a stable reduction in error within valid regions but suffer from sharp performance degradation in invalid regions—even when revisiting previously seen domains.

In contrast, the model adapted with AdaptBN progressively exhibit consistent improvement across both valid and invalid regions over time. Notably, when incorporating our dual-source dense pseudo-labels (denoted as *AT*), we observe that the error rate in invalid regions, which previously increased due to lack of supervision, is significantly mitigated. This indicates that the AdaptBN teacher provides reliable supervisory signals even in regions previously lacking ground-truth guidance, effectively supporting generalization across the entire image.

Table 4: Computational Cost Analysis. The average computational cost and error rate during 10 round experiments of DrivingStereo dataset.

| Method | Adapt. | #_Trainable (M) | Mem. (MB) | Runtime (ms) | Mean↓ D1-all (%) | Mean↓ EPE (px) |
|---|---|---|---|---|---|---|
| MADNet 2 | (a) no adapt. | - | 179 | 11 | 10.44 | 1.70 |
| | (b) FT | 3.22 | 276 | 31 | 6.13 | 1.73 |
| | (c) MAD++ | 3.22 | 398 | 18 | 5.86 | 1.41 |
| CoEx | (d) no adapt. | - | 694 | 23 | 5.56 | 1.26 |
| | (e) AdaptBN | 0.03 | 2704 | 139 | 3.08 | 0.94 |
| | (f) FT | 2.73 | 2744 | 255 | 3.04 | 0.92 |
| | (g) FT + EMA | 2.73 | 2835 | 267 | 2.96 | 0.92 |
| | (h) FT + AT | 2.79 | 5469 | 378 | 2.93 | 0.91 |
| **RobIA (ours)** | (i) AttEx-MoE | 1.19 | 1392 | 97.34 | 2.98 | 0.97 |
| | (j) AttEx-MoE + AT | 1.22 | 4096 | 204 | 2.77 | 0.91 |

Table 5: Ablation study on AttExMoE Architecture. The average D1-all and EPE error rate during 10 round experiments of DrivingStereo dataset.

| Router | Activation | Mean↓ D1-all | Mean↓ EPE |
|---|---|---|---|
| Shallow Embedding | ReLU | 3.16 | 0.96 |
| | Sigmoid | 3.11 | 0.96 |
| GAP | ReLU | 4.06 | 1.06 |
| | Sigmoid | 3.27 | 0.95 |
| Self-attention | ReLU | 3.61 | 1.13 |
| | Sigmoid | 3.09 | 0.94 |
| Column-wise Self-attention | ReLU | 4.11 | 1.07 |
| | Sigmoid | 3.20 | 0.97 |
| Row-wise Self-attention | ReLU | 3.77 | 1.09 |
| | Sigmoid (ours) | 2.98 | 0.97 |

**Computational Cost** Tab. 4 provides a comparative analysis of computational cost and adaptation performance. All runtime and memory measurements were recorded on an NVIDIA RTX 3090 GPU. MADNet2, while designed for test-time efficiency, shows consistent performance degradation under continual domain shifts, indicating limited robustness in dynamic settings. In contrast, our AttEx-MoE tuning (i) achieves comparable or superior accuracy with approximately half the number of trainable parameters and reduced memory usage compared to full model tuning methods such as (f) FT and (g) FT + EMA. (j) achieves the best overall mean D1-all and EPE, while maintaining a reasonable computational cost, demonstrating the effectiveness of input-aware expert selection for resource-efficient adaptation.

Table 6: Ablation study on $\lambda$.

| $\lambda$ | 1 dusky D1-all | EPE | 1 cloudy D1-all | EPE | 1 rainy D1-all | EPE | 10 dusky D1-all | EPE | 10 cloudy D1-all | EPE | 10 rainy D1-all | EPE | All↓ Mean D1-all | EPE |
|---|---|---|---|---|---|---|---|---|---|---|---|---|---|---|
| (a) 0.05 | 4.31 | 1.03 | 2.54 | 0.85 | 4.16 | 1.07 | 3.29 | 0.9 | 2.45 | 0.83 | 3.25 | 1.03 | 2.97 | 0.91 |
| (b) 0.1 | 4.28 | 1.03 | 2.4 | 0.84 | 4.54 | 1.16 | 2.4 | 0.84 | 2.24 | 0.82 | 3.02 | 1.00 | 2.77 | 0.91 |
| (c) 0.2 | 4.56 | 1.05 | 2.75 | 0.88 | 4.76 | 1.17 | 2.69 | 0.85 | 2.23 | 0.8 | 2.82 | 0.96 | 2.84 | 0.90 |
| (d) 0.3 | 4.69 | 1.07 | 2.88 | 0.89 | 5.39 | 1.27 | 2.44 | 0.83 | 2.14 | 0.8 | 2.84 | 0.95 | 2.89 | 0.91 |

## 5.3 Ablation Studies

**AttEx-MoE architecture.** Tab. 5 presents an ablation study on the architectural components of AttEx-MoE. We compare different gating network designs, including the type of router and the activation function (ReLU vs. Sigmoid), under the same training setting.

Shallow embedding and GAP-based routing result in higher error rates, supporting our observation that these methods lack sufficient spatial context for precise expert selection. Self-attention routers improve performance, but full 2D attention introduces additional computation without significant gains. Although column-wise self-attention does not leverage epipolar geometry, it partially preserves spatial structure and maintains comparable efficiency. However, its higher error rates further validate our design choice of row-wise attention. ReLU activation—commonly used to enforce sparsity—consistently underperforms across all router types, likely due to reduced expressivity under frozen backbones in the PEFT setting.

Our final design, combining row-wise self-attention with sigmoid-based expert excitation, achieves the best overall results. The router effectively captures structured global context along epipolar lines, and sigmoid activation enables more expressive and stable expert modulation. These results confirm the effectiveness of our Attend-and-Excite design for robust and efficient instance-aware adaptation.

**Loss weight $\lambda$ for $L_{teacher}$.** We ablated the loss weight $\lambda$ in Tab. 6. As shown in Tab. 6, we evaluated a range of $\lambda$ values over 10 CTTA rounds on the DrivingStereo sequence, using AttEx-MoE as the student model. The best performance is achieved with (b) $\lambda = 0.1$, which is also adopted in the main experimental results. Since the teacher model needs to newly adapt to the target domain and the effect of proxy supervision is relatively more important in the early stages, the larger $\lambda$ values tend to limit the model's adaptability at the early rounds of the adaptation. However, as teacher predictions become more accurate over time, the larger $\lambda$ leads to better performance in later rounds. In contrast, smaller values behave similarly to single-source supervision, resulting in higher error rates at the end of adaptation.

## 6 Conclusion

We presented **RobIA**, a robust and instance-aware framework for continual test-time adaptation (CTTA) in stereo depth estimation. RobIA addresses key challenges posed by dynamic domain shifts and sparse supervision through two core components: AttEx-MoE, a lightweight Mixture-of-Experts module guided by epipolar-aware self-attention, and a Robust AdaptBN Teacher that complements handcrafted pseudo-labels for generating dense supervision. This design enables flexible, input-specific adaptation while maintaining computational efficiency. Extensive experiments across dynamically shifting target domains demonstrate that RobIA consistently outperforms existing methods, highlighting the importance of instance-aware adaptation and hybrid supervision strategies for reliable deployment of stereo depth models in real-world settings.

**Limitations and Future Work.** While RobIA demonstrates strong performance in CTTA, it has certain limitations. Although AttEx-MoE offers input-aware adaptation, its reliance on predefined expert structures may limit flexibility in highly heterogeneous environments. Future work includes online expert refinement to further improve adaptation performance in long-term deployment scenarios.

## Acknowledgements

This work was supported by IITP grant funded by MSIT (No. RS-2022-00155966: AI Convergence Innovation Human Resources Development (Ewha Womans University) and No. RS-2021-II212068: AI Innovation Hub).

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

# A  Implementation Details

For the warm-up process, we trained the model using the Adam optimizer for 10 epochs with a fixed learning rate of 5e-4. During test-time adaptation (TTA), we also used Adam optimizer across all methods. The learning rate was set to 5e-6 for training the AdaptBN teacher model and 1e-5 for MADNet2 [45]. Tab. 7 reports the learning rates of student models used in experiments. For full-tuning methods, we used relatively smaller learning rates—approximately 10 times lower than those used in efficient tuning methods such as AdaptBN, MetaNet [12], and our AttEx-MoE. This choice is motivated by the observation that large learning rates in full-tuning settings significantly impair generalization performance.

Our baseline model is CoEx [43], a compact and real-time stereo matching network. The feature extractor in CoEx is composed of four scale levels, with upsampling modules built using long skip connections at each scale. To implement EcoTTA [12] in the CTTA setting for stereo matching , we inserted its meta network into the MobileNetV2 backbone of CoEx at four scale-level blocks (K=4).

Table 7: Learning rates.

| Dataset | Exp. | Learning Rate |
|---|---|---|
| Full tuning | | |
| DrivingStereo | CTTA | 5e-7 |
| DSEC | CTTA | 2e-6 |
| KITTI RAW | CTTA | 1e-5 |
| Efficient tuning | | |
| DrivingStereo | CTTA | 5e-6 |
| DSEC | CTTA | 3e-5 |
| KITTI RAW | CTTA | 1e-4 |

# B  Additional Analysis and Ablation Studies

**Pseudo-label visualization.**  Fig. 3 visualizes the pseudo-labels generated by different methods after ten rounds of adaptation from the rainy sequences of the DrivingStereo dataset. On each, we reported disparity maps of the student model and the corresponding error rate. Note that, for sparse pseudo-labels, we only measured error for regions where both the pseudo-label and the ground truth are valid.

The handcrafted sparse pseudo-labels often lack supervisory signals in challenging regions for stereo matching, such as reflective surface, low-texture regions, or occlusions. Furthermore, this issue extends to image borders, where the hand-crafted stereo algorithms (*e.g.*, SGM) often struggles due

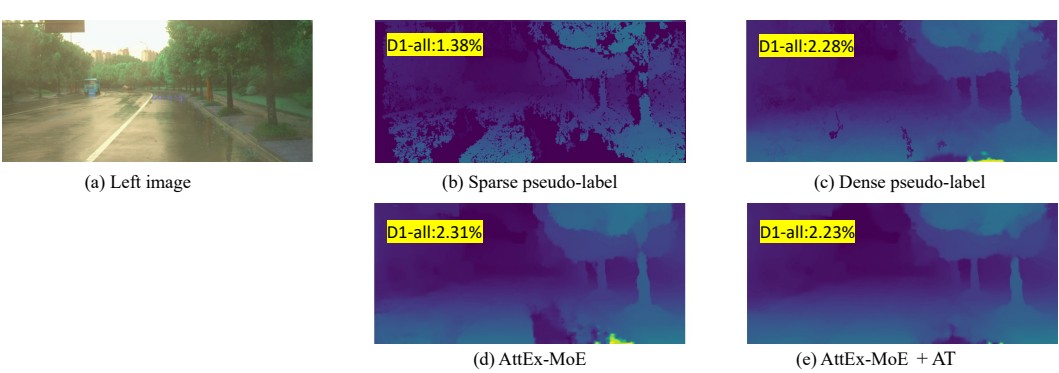

|  |  |  |
|---|---|---|
| (a) Left image | (b) Sparse pseudo-label | (c) Dense pseudo-label |
| | (d) AttEx-MoE | (e) AttEx-MoE + AT |

Figure 3: Pseudo-labels (top row) and predictions (bottom row) after ten adaptation rounds. We visualize the sparse handcrafted pseudo-label (b) and the dense pseudo-label using the AdaptBN teacher (c), and the student predictions of AttEx-MoE trained with sparse (d) and dense (e) supervision. (a) shows the input left image.

Table 8: Ablation study on Pseudo Supervision. Average D1-all and EPE over 10 adaptation rounds on DrivingStereo. We use single- or dual-source supervision based on whether only one or both of handcrafted and learned signals are used.

| Source | Method | 1 dusky D1-all | 1 dusky EPE | 1 cloudy D1-all | 1 cloudy EPE | 1 rainy D1-all | 1 rainy EPE | 5 dusky D1-all | 5 dusky EPE | 5 cloudy D1-all | 5 cloudy EPE | 5 rainy D1-all | 5 rainy EPE | 10 dusky D1-all | 10 dusky EPE | 10 cloudy D1-all | 10 cloudy EPE | 10 rainy D1-all | 10 rainy EPE | All↓ Mean D1-all | All↓ Mean EPE |
|---|---|---|---|---|---|---|---|---|---|---|---|---|---|---|---|---|---|---|---|---|---|
| Single | (a) proxy | 4.01 | 1.01 | 2.4 | 0.84 | 4.44 | 1.13 | 2.54 | 0.86 | 2.23 | 0.83 | 3.89 | 1.2 | 2.72 | 0.88 | 2.29 | 0.84 | 3.89 | 1.22 | 2.98 | 0.97 |
| | (b) photometric | 4.3 | 1.04 | 2.45 | 0.85 | 4.76 | 1.18 | 3.21 | 0.95 | 2.75 | 0.89 | 4.74 | 1.25 | 3.67 | 1.01 | 2.91 | 0.91 | 4.45 | 1.22 | 3.55 | 1.02 |
| | (c) AdaptBN | 5.5 | 1.13 | 3.48 | 0.98 | 6.91 | 1.5 | 3.73 | 0.97 | 2.83 | 0.88 | 5.65 | 1.3 | 3.15 | 0.92 | 2.55 | 0.85 | 5.03 | 1.21 | 4.16 | 1.06 |
| Dual | proxy | | | | | | | | | | | | | | | | | | | | |
| | + (d) Source | 4.28 | 1.03 | 2.4 | 0.84 | 4.62 | 1.18 | 2.75 | 0.88 | 2.3 | 0.82 | 4.0 | 1.13 | 2.78 | 0.87 | 2.31 | 0.82 | 3.82 | 1.1 | 3.09 | 0.95 |
| | + (e) EMA | 4.27 | 1.03 | 2.38 | 0.84 | 4.56 | 1.16 | 2.84 | 0.88 | 2.26 | 0.82 | 4.09 | 1.16 | 3.11 | 0.91 | 2.32 | 0.82 | 4.03 | 1.15 | 3.16 | 0.96 |
| | + (f) photometric [46] | 4.02 | 1.01 | 2.4 | 0.84 | 4.44 | 1.13 | 2.54 | 0.86 | 2.23 | 0.83 | 3.87 | 1.18 | 2.69 | 0.87 | 2.29 | 0.84 | 3.87 | 1.22 | 2.98 | 0.96 |
| | + (g) **AT (ours)** | 4.28 | 1.03 | 2.4 | 0.84 | 4.54 | 1.16 | 2.46 | 0.85 | 2.2 | 0.81 | 3.27 | 1.01 | 2.4 | 0.84 | 2.24 | 0.82 | 3.02 | 1.0 | 2.77 | 0.91 |

to the lack of neighboring pixels for reliable cost aggregation. As a result, models relying solely on these handcrafted labels tend to exhibit reduced adaptability in CTTA. In contrast, our dense pseudo-labels provide complete spatial coverage, allowing the model to adapt effectively even in unreliable regions. This confirms the role of dense, teacher-guided pseudo-supervision in enhancing spatial robustness under continual adaptation settings.

**Pseudo-supervision ablation.** Tab. 8 presents an ablation study comparing various pseudo-supervision strategies using our AttEx-MoE based Parameter-efficient tuning method. Single-source supervision methods (a–c) perform worse overall. (a) Proxy supervises the model effectively, but the limited guidance of sparse handcrafted pseudo-supervision means that the model's performance degradation later in the round. (b) photometric loss leads to increasingly higher error rates due to noisy guidance. (c) AdaptBN supervision alone shows more stable error reduction but underperforms when used without proxy supervision, as the teacher model itself requires time to adapt to the target domain.

Dual-source variants (d–g) combine proxy labels with additional supervision to improve generalization across the entire input space. Although using a fixed source model (d) enhances stability, it lacks adaptability under distribution shift. (e) EMA initially helps, but tends to propagate student errors. Following [46], (f) augments proxy label with photometric and smoothing losses to compensate for its sparsity, but does not yield meaningful improvement over (a) proxy-only. In contrast, our method (g) leverages a robust AdaptBN teacher to provide dense supervision, achieving the best overall results (D1-all 2.77% , EPE 0.91), maintaining the performance in long-term and continuously changing conditions. These findings underscore the importance of dense, complementary supervision in overcoming the limitations of sparse pseudo-labels during continual adaptation.

## C  Additional Experiments Results

Table 9: Performance comparison of Continual Test-time Adaptation on DrivingStereo benchmark over 10 rounds for IGEV-Stereo and LightStereo backbone. **Bold** denotes best and *AT* denotes our method with dense pseudo-label $D_{\text{teacher}}$.

| Backbone | Adapt. | Mean↓ DrivingStereo D1-all (%) | Mean↓ DrivingStereo EPE (px) |
|---|---|---|---|
| IGEV-Stereo | (a) no adapt. | 7.26 | 1.32 |
| | (b) AdaptBN | 3.87 | 0.98 |
| | (c) FT | 3.56 | 0.96 |
| | (d) AttEx-MoE | 3.15 | 0.92 |
| | (e) AttEx-MoE + AT | **2.76** | **0.87** |
| LightStereo | (a) no adapt. | 18.59 | 3.11 |
| | (b) AdaptBN | 5.95 | 1.74 |
| | (c) FT | 5.91 | 1.72 |
| | (d) AttEx-MoE | 5.76 | 1.28 |
| | (f) AttEx-MoE + AT | **4.95** | **1.11** |

Table 10: Performance comparison of Continual Test-time Adaptation on DrivingStereo and KITTI RAW benchmark over 10 rounds for MADNet2 backbone. **Bold** denotes best and *AT* denotes our method with dense pseudo-label $D_{\text{teacher}}$.

| Backbone | Adapt. | Mean↓ | | | | Runtime |
| | | DrivingStereo | | KITTI | | |
| | | D1-all (%) | EPE (px) | D1-all (%) | EPE (px) | (ms) |
|---|---|---|---|---|---|---|
| MADNet2 | (a) no adapt. | 10.44 | 1.70 | 3.92 | 1.09 | 11 |
| | (b) FT | 6.13 | 1.73 | 1.13 | **0.85** | 31 |
| | (c) MAD++ | 5.86 | 1.41 | 1.24 | 0.88 | 18 |
| | (d) AttEx-MoE | 5.53 | 1.20 | 1.12 | 0.86 | 23 |
| | (e) AttEx-MoE + AT | **4.83** | **1.09** | **1.10** | **0.85** | 49 |

As shown in Tab. 9, beyond the 3D cost aggregation model CoEx, we also evaluated our method on *IGEV-Stereo* [47], a widely used iterative-refinement backbone, and on *LightStereo* [48], a lightweight real-time network with 2D cost aggregation. On all three architectures, our RobIA with AttEx-MoE and AdaptBN-Teacher consistently improves accuracy, demonstrating that our approach generalizes well to various stereo architectures.

Our components remain consistently effective when applied to IGEV-Stereo. Without adaptation, (a) IGEV-Source suffers from significant domain shift, yielding high D1-all errors 7.26% and EPE 1.32. While tuning only BN parameters ((b) AdaptBN) or full fine-tuning ((c) FULL++) reduces error, both approaches still struggle (Da-all (b) 3.87%, (c) 3.56%). In contrast, (d) our AttEx-MoE based PEFT lowers D1-all to 3.15%, and our RobIA implementation ((e) AttEx-MoE + AT) further improves it to 2.76%. This demonstrates that our plug-and-play modules effectively enhance generalization even on top-performing backbones.

Tab. 9 also shows that RobIA generalizes well to LightStereo. Without adaptation, (a) LightStereo-Source performs poorly due to severe domain shift (18.59%). (b) AdaptBN alone improves results to 5.95%, yet falls short on dynamic scenes. Our AttEx-MoE module (e) further reduces the D1-all error to 5.76%, and RobIA (f) achieves the best performance (D1-all 4.95%, EPE 1.11), confirming the robustness and plug-and-play nature of our method even on compact backbones. These results with IGEV-Stereo and LightStereo demonstrate that RobIA is effective across both strong and lightweight backbones, and maintains robustness under significant domain shifts.

Furthermore, we re-implemented our method, including AttEx-MoE and the AdaptBN teacher, on top of the *MADNet2* [8] encoder, which is also used by MAD++. Since MADNet2 lacks normalization layers, we inserted BatchNorm layers after each convolution to enable AdaptBN and maintain architectural consistency.

As shown in Table 10, on KITTI, which presents relatively mild domain shifts, (b) FULL++ achieves D1-all 1.13% by updating all layers, while (d) our AttEx-MoE method achieves a better result (1.12%) with fewer parameters. (e) RobIA, which combines AttEx-MoE with the AdaptBN teacher, maintains a strong KITTI score of 1.1%. On DrivingStereo, which exhibits stronger domain shifts, (b) FULL++ struggles with large domain shifts (6.13%), and (c) MAD++ provides only minor improvement (5.86%). In contrast, (d) our AttEx-MoE method further reduces the error to 5.53%, and (e) RobIA achieves the best result at 4.83%. These results show that our input-aware AttEx-MoE gate, combined with dual-source supervision, matches full tuning on stable domains, and significantly outperforms both full and modular adaptation approaches under large distribution shifts.

Table 11: Performance comparison for Sequential Continual Test-time Adaptation across different datasets, from KITTI RAW (rounds 1–5) to DrivingStereo (rounds 6–10). **Bold** denotes best and *AT* denotes our method with dense pseudo-label $D_{\text{teacher}}$.

| Backbone | Adapt. | KITTI Round 1 → 5 | | DrivingStereo (after KITTI) Round 6 → 10 | | ALL | |
| | | D1-all (%) | EPE (px) | D1-all (%) | EPE (px) | D1-all (%) | EPE (px) |
|---|---|---|---|---|---|---|---|
| CoEx | (a) AdaptBN | 1.11 | 0.84 | 5.90 | 1.70 | 3.16 | 1.21 |
| | (b) AttEx-MoE | 1.11 | 0.85 | 4.22 | 1.08 | 2.44 | 0.95 |
| | (c) AttEx-MoE + AT | 1.15 | 0.84 | 2.66 | 0.90 | **1.80** | **0.87** |

Table 12: Performance comparison of Test-time Adaptation on **KITTI RAW** benchmarks. **Bold** denotes best and *AT* denotes our method with dense pseudo-label $D_{\text{teacher}}$.

| | | City (8027 frames) | | Residential (28067 frames) | | Campus$^{(\times 2)}$ (1149 × 2 frames) | | Road (5674 frames) | | Mean | |
|---|---|---|---|---|---|---|---|---|---|---|---|
| Method | Condition Adapt. | D1-all | EPE | D1-all | EPE | D1-all | EPE | D1-all | EPE | D1-all | EPE |
| RAFT-Stereo [49] | (a) no adapt. | 1.55 | 0.89 | 1.77 | 0.82 | 2.53 | 0.89 | 1.77 | 0.85 | 1.90 | 0.86 |
| CREStereo [50] | (b) no adapt. | 1.87 | 0.99 | 1.71 | 0.89 | 3.21 | 1.07 | 2.00 | 0.89 | 2.20 | 0.96 |
| IGEV-Stereo [47] | (c) no adapt. | 2.26 | 1.00 | 2.56 | 0.94 | 3.01 | 0.99 | 2.52 | 0.96 | 2.58 | 0.97 |
| UniMatch [51] | (d) no adapt. | 2.66 | 1.13 | 3.20 | 1.10 | 3.10 | 1.13 | 2.26 | 1.08 | 2.81 | 1.11 |
| MADNet 2 [45] | (e) no adapt. | 4.04 | 1.10 | 4.05 | 1.03 | 6.07 | 1.29 | 4.01 | 1.08 | 4.54 | 1.13 |
| | (f) FT | 1.23 | 0.90 | 1.05 | 0.80 | 2.39 | 0.92 | 1.02 | 0.83 | 1.42 | 0.86 |
| | (g) MAD++ | 1.39 | 0.93 | 1.16 | 0.83 | 2.88 | 1.00 | 1.14 | 0.85 | 1.64 | 0.90 |
| CoEx [43] | (h) no adapt. | 2.66 | 1.07 | 2.66 | 0.99 | 3.65 | 1.11 | 2.46 | 0.98 | 2.86 | 1.04 |
| | (i) FT | 0.83 | **0.84** | 0.75 | 0.76 | 1.49 | **0.8** | **0.75** | **0.79** | 0.96 | 0.80 |
| | (j) FT + AT | **0.8** | 0.84 | **0.66** | **0.74** | **1.45** | 0.8 | 0.79 | 0.8 | **0.93** | **0.80** |
| **RobIA (ours)** | (k) AttEx-MoE | 0.99 | 0.87 | 0.96 | 0.78 | 1.6 | 0.82 | 0.9 | 0.81 | 1.11 | 0.82 |
| | (l) AttEx-MoE + AT | 1.05 | 0.87 | 0.91 | 0.78 | 1.56 | 0.81 | 0.94 | 0.83 | 1.12 | 0.82 |

We simulated a sequential CTTA scenario where the model first adapts to KITTI for 5 rounds, followed by adaptation to DrivingStereo for another 5 rounds, mimicking a realistic progression from a stable to a more challenging domain. Results are shown in Tab. 11. (b) and (c) show similar error rates with (a) on KITTI, but they achieve substantially lower error on DrivingStereo. (a), despite strong performance of 1.11% on KITTI, fails to adapt in the second phase (5.90%). These results highlight the robustness of our AttEx-MoE with AdaptBN teacher, which generalizes better under sequential, cross-domain conditions, mirroring realistic deployment settings.

**TTA Experiments.** Previous studies have demonstrated effective performance under TTA, which motivates us to assess whether our approach, designed for CTTA, also generalizes well in this setting. To evaluate the effectiveness of our method under standard test-time adaptation (TTA) settings, we conducted experiments on three real-world stereo datasets: KITTI RAW, DrivingStereo, and DSEC. The results are reported in Tab. 12, Tab. 13, and Tab. 14 for KITTI RAW, DrivingStereo, and DSEC, respectively. Following prior work [45], we compared against MADNet2 [45], and several state-of-the-art stereo models known for strong generalization performance. The stereo models only trained on synthetic source datasets (a–d) exhibit significant performance degradation on target domains due to domain shifts. While adaptation-based methods generally improve accuracy, performance gains remain limited for efficiency-oriented state-of-the-art TTA methods (e–g).

Our efficient tuning method (k), which employs AttEx-MoE with input-aware feature excitation, achieves higher accuracy than prior TTA baselines, while remaining comparable to full model tuning and reducing computational cost, as discussed in the main paper. Results supervised by our dense pseudo-labels are shown in (j) and (l). Notably, substantial improvements by dense supervisions with AdaptBN Teacher are observed on DrivingStereo and DSEC, where pseudo-label sparsity is higher due to challenging conditions including adverse weather and nighttime imagery.

These findings demonstrate that our method is not only effective in the proposed continual adaptation scenario but also exhibits effective adaptability in standard TTA settings involving long-term adaptation within individual domains.

**Qualitative Comparison of CTTA Results.** We report qualitative comparisons of disparity maps predicted by various models and supervision strategies evaluated in our CTTA experiments. Figs. 4 and 5 show examples from the cloudy and rainy sequences of the DrivingStereo dataset. For each example, predictions are visualized at round 1, 5, and 10 to highlight temporal adaptation behaviors.

We observed that our method consistently improves predictions across rounds, particularly in challenging regions such as reflective surfaces, low-texture areas, and image borders. Models trained with only sparse supervision show limited generalization beyond the confident regions of the handcrafted pseudo-labels, which limits their adaptability in less confident areas over time. In contrast, dense pseudo-supervision enables broader coverage and leads to stable improvements across the entire image, demonstrating stronger generalization under continual domain shifts.

Table 13: Performance comparison of Test-time Adaptation on **DrivingStereo** benchmarks. **Bold** denotes best and *AT* denotes our method with dense pseudo-label $D_{\text{teacher}}$.

| Method | Condition Adapt. | (1667 frames) dusky D1-all | EPE | (1119 frames) cloudy D1-all | EPE | (4950 frames) rainy D1-all | EPE | Mean D1-all | EPE |
|---|---|---|---|---|---|---|---|---|---|
| RAFT-Stereo [49] | (a) no adapt. | 11.52 | 1.59 | 3.08 | 0.88 | 4.18 | 1.02 | 6.26 | 1.16 |
| CREStereo [50] | (b) no adapt. | 17.43 | 3.61 | 7.08 | 1.23 | 4.08 | 1.07 | 9.53 | 1.97 |
| IGEV-Stereo [47] | (c) no adapt. | 11.70 | 1.85 | 3.57 | 0.95 | 5.27 | 1.26 | 6.95 | 1.35 |
| UniMatch [51] | (d) no adapt. | 14.84 | 2.69 | 7.51 | 1.27 | 5.78 | 1.25 | 9.38 | 1.74 |
| MADNet 2 [45] | (e) no adapt. | 16.47 | 3.03 | 13.16 | 1.66 | 6.72 | 1.35 | 12.12 | 2.01 |
|  | (f) FT | 10.34 | 2.27 | 4.41 | 1.04 | 5.20 | 1.63 | 6.65 | 1.65 |
|  | (g) MAD++ | 10.06 | **2.01** | 5.25 | 1.09 | 4.34 | 1.09 | 6.55 | 1.40 |
| CoEx [43] | (h) no adapt. | 13.55 | 3.02 | 5.24 | 1.12 | 4.12 | 1.15 | 7.64 | 1.76 |
| CoEx | (i) FT | 8.85 | 2.29 | 3.04 | 0.9 | 3.7 | 1.27 | 5.20 | 1.49 |
|  | (j) FT + AT | **7.93** | 2.09 | 2.63 | **0.88** | **2.29** | **0.84** | **4.28** | **1.27** |
| **RobIA (ours)** | (k) AttEx-MoE | 9.05 | 2.28 | 3.21 | 0.96 | 2.8 | 0.91 | 5.02 | 1.38 |
|  | (l) AttEx-MoE + AT | 8.27 | 2.18 | **2.61** | 0.91 | 2.77 | 0.92 | 4.55 | 1.34 |

Table 14: Performance comparison of Test-time Adaptation on **DSEC** benchmarks. **Bold** denotes best and *AT* denotes our method with dense pseudo-label $D_{\text{teacher}}$.

| Method | Condition Adapt. | (883 frames) Night#1 D1-all | EPE | (1813 frames) Night#2 D1-all | EPE | (2315 frames) Night#3 D1-all | EPE | (2405 frames) Night#4 D1-all | EPE | Mean D1-all | EPE |
|---|---|---|---|---|---|---|---|---|---|---|---|
| RAFT-Stereo [49] | (a) no adapt. | 13.04 | 3.41 | 21.64 | 4.26 | 10.91 | 1.91 | 10.07 | 1.68 | 13.92 | 2.82 |
| CREStereo [50] | (b) no adapt. | 11.34 | 2.38 | 23.48 | 3.19 | 15.37 | 2.39 | 12.42 | 1.75 | 15.65 | 2.43 |
| IGEV-Stereo [47] | (c) no adapt. | 9.14 | 1.85 | 11.97 | 1.96 | 12.65 | 2.01 | 10.01 | 1.66 | 10.94 | 1.87 |
| UniMatch [51] | (d) no adapt. | 34.29 | 5.43 | 39.80 | 5.32 | 26.75 | 3.29 | 26.29 | 3.28 | 31.78 | 4.33 |
| MADNet 2 [45] | (e) no adapt. | 8.94 | 1.97 | 13.86 | 2.32 | 10.63 | 1.83 | 10.55 | 1.69 | 11.00 | 1.95 |
|  | (f) FT | 4.69 | 1.28 | 7.13 | 1.43 | 6.20 | 1.35 | 6.06 | 1.27 | 6.02 | 1.33 |
|  | (g) MAD++ | 5.66 | 1.43 | 8.39 | 1.53 | 7.91 | 1.50 | 7.79 | 1.39 | 7.44 | 1.46 |
| CoEx [43] | (h) no adapt. | 6.14 | 1.52 | 10.27 | 1.77 | 7.82 | 1.58 | 7.64 | 1.45 | 7.97 | 1.58 |
|  | (i) FT | 3.55 | 1.1 | 5.14 | 1.2 | 4.39 | 1.11 | 4.41 | 1.07 | 4.37 | 1.12 |
|  | (j) FT + AT | 3.65 | 1.1 | **4.98** | **1.18** | **4.19** | **1.1** | **4.12** | **1.04** | **4.24** | **1.11** |
| **RobIA (ours)** | (k) AttEx-MoE | 3.66 | 1.13 | 5.32 | 1.23 | 4.59 | 1.15 | 4.85 | 1.13 | 4.61 | 1.16 |
|  | (l) AttEx-MoE + AT | **3.51** | **1.09** | 5.38 | 1.23 | 4.77 | 1.18 | 4.58 | 1.1 | 4.56 | 1.15 |

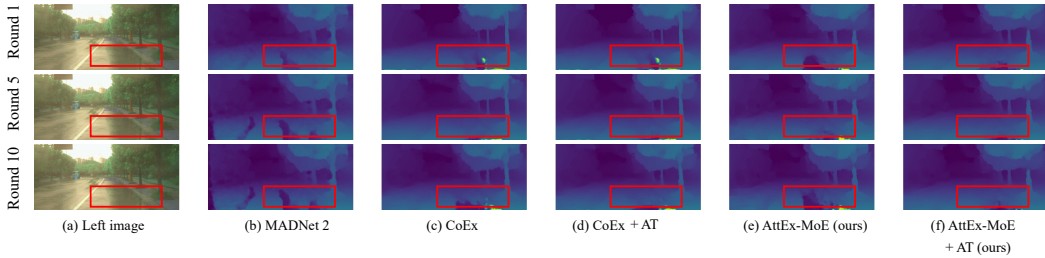

Figure 4: Qualitative results for cloudy sequences in the DrivingStereo dataset.

Figure 5: Qualitative results for rainy sequences in the DrivingStereo dataset.

