# OpenReview forum: "RobIA: Robust Instance-aware Continual Test-time Adaptation for Deep Stereo"
_NeurIPS.cc/2025/Conference — NeurIPS 2025 poster_

### Official Review · Reviewer_VsX9 · 2025-06-24

**Clarity:** 4
**Significance:** 3
**Originality:** 3
**Rating:** 5
**Confidence:** 4

**Summary:**

The article introduces a novel method for Continual Test-Time Adaptation in Deep Stereo. The paper's main contributions are: a new Mixture of Experts-based module for convolutional networks, and a new strategy for generating pseudo-labels during training.
Regarding the first contribution, the proposed module uses row-wise self-attention to extract an embedding that is fed into a gating network. This approach is both efficient and preserves some spatial structure from the input images. The resulting embedding is used to produce channel-wise modulations for each training instance, enabling greater model flexibility. Moreover, using a sigmoid activation function instead of ReLU helps retain information across all channels of the base model, which is particularly beneficial in scenarios where the backbone is frozen.
As for the second contribution, the method proposes generating dense pseudo-labels by combining sparse pseudo-labels obtained through traditional methods (e.g., SGM) with dense pseudo-labels generated by a teacher network for the remaining regions. Specifically, the teacher network is implemented using an AdaptBN version of the base model, rather than a conventional EMA approach. This choice is motivated by the intuition that AdaptBN is less prone to reinforcing the errors of hand-crafted pseudo-labels, thus providing a more reliable supervision signal.

**Questions:**

See Weaknesses section above.

**Ethical Concerns:**

["NO or VERY MINOR ethics concerns only"]

**Final Justification:**

After carefully reviewing the rebuttal, some of my concerns, particularly points B, C, and D, remain only partially addressed. However, given the authors’ commitment to expand on these aspects in the camera-ready version, and considering the overall strength of the paper, I recommend acceptance.

**Limitations:**

Some limitations are presented in the paper.

**Paper Formatting Concerns:**

None.

**Quality:**

3

**Strengths And Weaknesses:**

__Strength__

- The paper is well-written, well-structured, and well-curated.

- The various ideas have been thoroughly motivated, both logically and experimentally.

- The experiments are rigorously conducted and also well-described in the paper.

__Weaknesses__

In order of importance:

A. Some stereo networks have architectural differences compared to CoeX. For instance, could the proposed method be applied to frameworks such as HITNet [a] or TemporalStereo [b]?

[a] Tankovich et al. HITNet: Hierarchical Iterative Tile Refinement Network for Real-time Stereo Matching. CVPR 2021.

[b] Zhang et al. TemporalStereo - Efficient Spatial-Temporal Stereo Matching Network. IROS 2023

B. The AttEX-MoE strategy appears to be particularly tailored to CNN-based networks. However, many recent stereo architectures rely on transformer-based approaches. Is the proposed method applicable to these types of architectures as well?

C. The pseudo-label generation mechanism using AdaptBN implicitly assumes that the network employs Batch Normalization (BN) layers. How could this technique be extended to scenarios where the base network uses a different type of normalization?

D. While row-wise self-attentions improve computational efficiency, the emphasis on epipolar lines is unclear. If leveraging epipolar geometry to estimate embeddings is a key factor for performance, this should be validated experimentally. For example, one possible approach could be to compare it with a strategy that does not exploit epipolar geometry but still partially preserves spatial structure and offers similar efficiency (e.g., using column-wise instead of row-wise embeddings).


__Minor Weaknesses__

a.	Missing bolds and underlines to highlight the best results in Tables 2, 3, and 4

b.	Several highly relevant results are only presented in the appendix (e.g., Tables 8, 9, and 10). While it's understandable that these results were moved to the appendix due to space constraints, they should at least be referenced or briefly discussed in the main paper to emphasize their importance.

__Overall Comments__

The paper is well-organized and structured. It presents novel ideas that are well-motivated and interesting. The main limitation of the work lies in its specificity to a particular architecture, relying on specific normalization layers and a convolution-based design. A better understanding of this limitation could lead me to provide a more grounded result. It would also be interesting to investigate whether the emphasis on epipolar geometry is necessary for the AttEX-MoE component.

---

> ### Author Rebuttal · Authors · 2025-07-31
>
> We appreciate the reviewer **VsX9** for recognizing the significance of our CTTA framework for stereo depth estimation, the motivation behind our MoE and AdaptBN-based components, and the clarity and thoroughness of the manuscript and experiments.
>
>
> ---
>
> ## Weakness A. Applicability to Other Architectures
>
> Yes, our method is applicable to frameworks such as HITNet [A] and TemporalStereo [B], as both AttEx-MoE and AdaptBN-Teacher are designed as plug-and-play modules compatible with a wide range of CNN-based stereo architectures.
>
> AttEx-MoE applies instance-aware, channel-wise excitation exclusively within the feature extractor, as this stage is critical for capturing high-quality, geometry-aware features for stereo matching.
> Its row-wise self-attention router operates along epipolar lines and naturally aligns with the geometry of rectified stereo images, making it broadly applicable to architectures that process stereo pairs. We elaborate further on this benefit in the **Weakness D. Why Epipolar Line?** section.
>
> Similarly, AdaptBN is applicable to any CNN networks using batch normalization. We provide further discussion in the **Weakness C. Substitutes for Batch Normalization** section.
>
>
> ---
>
> ## Weakness B. Applicability to Transformer-based Architectures
>
> Yes, AttEx-MoE is compatible with Transformer-based stereo architectures.
>
> While our experiments focus on CNN backbones (e.g., CoEx), AttEx-MoE is backbone-agnostic. Its components, the row-wise self-attention router and Sigmoid gating, operate on feature maps, and can be applied to any encoder as long as it maintains image's spatial feature structure.
>
> In Transformer-based stereo models [C] [D], token sequences can be reshaped into feature maps, enabling seamless integration of AttEx-MoE into the encoder without architectural changes.
>
>
> ---
>
> ## Weakness C. Substitutes for Batch Normalization
>
> AdaptBN is not inherently tied to BatchNorm but is based on adapting a small set of affine parameters within a frozen backbone. While we do not experiment with other normalization schemes in this work, the underlying principle is conceptually generalizable:
>
> * **GroupNorm / InstanceNorm**: Like BN, both provide per-channel affine parameters $(\gamma, \beta)$, suggesting that similar adaptation (*e.g.*, AdaptGN or AdaptIN) could be feasible.
>
> * **LayerNorm (e.g., in ViT)**: Despite operating over hidden dimensions, its affine parameters $(\gamma_{LN}, \beta_{LN})$ may also be adapted per layer to enable token-level adaptation.
>
> For backbones without explicit affine layers (*e.g.*, Conv + ReLU), a lightweight 1×1 'scale-shift' adapter (two parameters per channel) can be inserted with minimal overhead.
>
> In summary, AdaptBN’s effectiveness stems from adapting affine modulation parameters, not BatchNorm specifically. This approach may be extended to other architectures with compatible parameterizations. We will clarify this in the revised manuscript.
>
>
> ---
>
> ## Weakness D. Why Epipolar Line?
>
> While full self-attention is effective for modeling global dependencies, it incurs prohibitive computational cost. In rectified stereo geometry, corresponding points across view lie along the same image row, making horizontal context more relevant than vertical context for disparity estimation [C].
> We thereby adopt row-wise self-attention, which focuses along epipolar-consistent scanlines. This preserves disparity-relevant context while reducing the attention complexity from $H \times W \times H \times W$ to $H \times W \times W$, lowering memory and FLOPs by a factor of $H$.
> Thus, row-wise attention is not merely a speed heuristic, but a geometry-aware inductive bias that enhances both efficiency and performance.
> Empirical results in Tab. 4 support this design: among various router alternatives, row-wise attention achieves the best performance.
>
> ---
> ## About Paper Formatting Concerns
>
> We appreciate your careful reading and pointing out the formatting issues. We will fix the boldface formatting in Table 1 and the other tables.
>
> ---
>
> ## Citations
> * [A] Tankovich, Vladimir, et al. "Hitnet: Hierarchical iterative tile refinement network for real-time stereo matching." Proceedings of the IEEE/CVF conference on computer vision and pattern recognition. 2021.
> * [B] Zhang, Youmin, Matteo Poggi, and Stefano Mattoccia. "Temporalstereo: Efficient spatial-temporal stereo matching network." 2023 IEEE/RSJ International Conference on Intelligent Robots and Systems (IROS). IEEE, 2023.
> * [C] Li, Zhaoshuo, et al. "Revisiting stereo depth estimation from a sequence-to-sequence perspective with transformers." Proceedings of the IEEE/CVF international conference on computer vision. 2021.
> * [D] Weinzaepfel, Philippe, et al. "Croco v2: Improved cross-view completion pre-training for stereo matching and optical flow." Proceedings of the IEEE/CVF International Conference on Computer Vision. 2023.

---

> > ### Comment · Reviewer_VsX9 · 2025-08-01
> >
> > After carefully reviewing the rebuttal, some of my concerns remain.
> >
> > B. and C. I fully acknowledge that the method is, in principle, applicable to a range of architectures, and that its formulation could be made more general. Indeed, throughout the paper — starting from the title — the method is presented as general and independent of the underlying architecture. This is precisely the origin of my concern: if the method is meant to be general, how effective is it when applied to substantially different architectures (e.g., transformers) with different types of layers (e.g., no batch normalization)? The current experiments validate its effectiveness only within a specific subset of architectures with particular characteristics.
> >
> > D. Similarly, while I understand the theoretical motivation for leveraging epipolar geometry, and while the efficiency improvements are indeed compelling, in my opinion, the experiments in Table 4 do not clearly demonstrate that using rows yields consistently better performance. In fact, the improvements appear to be marginal and limited to a single metric. Including an experiment based on columns would have helped to ascertain whether the choice of rows is not merely an elegant narrative, but a truly essential design decision grounded in empirical evidence.
> >
> > That said, I continue to regard the paper as solid overall, and I therefore maintain my initial borderline accept recommendation after the rebuttal.

---

> > > ### Author Response · Authors · 2025-08-06
> > >
> > > Thank you for the additional feedback. We appreciate the opportunity to clarify the scope of the current submission and how we plan to address the remaining concerns.
> > >
> > > **B, C.** We agree that evaluating transformer-based stereo backbones would further support the generality of our method. While the proposed design is architecture-agnostic in principle, we intentionally focus our experiments on diverse CNN-based stereo networks to align with our core motivation: revisiting the under-explored yet practically important setting of *efficient, CNN-compatible mixture-of-experts (MoE) design* for continual test-time adaptation (CTTA).
> > >
> > > In particular, for MADNet2—which was originally designed without normalization layers—we inserted BatchNorm layers and trained only the affine parameters, resulting in improved accuracy (see **Table A** in the rebuttal to Reviewer **4q4z**). While this does not fully address the broader class of architectures raised in the review, it does provide a concrete example of our method’s robustness under minimal normalization and limited architectural modification.
> > >
> > > We will clarify this design choice and its motivation more explicitly in the final manuscript and plan to extend the evaluation to transformer-based backbones (e.g., STTR, CroCo) in future work.
> > >
> > > **D.** Thank you for the suggestion. We agree that including a column-wise attention baseline would further reinforce the design motivation. Although it was not included due to time constraints, we plan to incorporate it in the camera-ready version.
> > >
> > >
> > > We sincerely thank the reviewer for the thoughtful and constructive feedback throughout the review process, and we truly appreciate the decision to recommend acceptance of our paper.

---

> > > > ### Comment · Reviewer_VsX9 · 2025-08-07
> > > >
> > > > Thank you for your response. I will recommend acceptance of the paper. I look forward to seeing the experimental results included in the camera-ready version.

---

### Official Review · Reviewer_4p4z · 2025-06-29

**Clarity:** 3
**Significance:** 2
**Originality:** 3
**Rating:** 4
**Confidence:** 3

**Summary:**

This paper proposes **RobIA**, a novel Robust, Instance-Aware framework for Continual Test-Time Adaptation (CTTA) in stereo depth estimation. RobIA integrates two key components: (1) Attend-and-Excite Mixture-of-Experts (AttEx-MoE), a parameter-efficient module that dynamically routes input to frozen experts via lightweight self-attention mechanism tailored to epipolar geometry, and (2) Robust AdaptBN Teacher, a PEFT-based teacher model that provides dense pseudo-supervision by complementing sparse handcrafted labels. Extensive experiments demonstrate that RobIA achieves superior adaptation performance across dynamic target domains while maintaining computational efficiency.

**Questions:**

Please refer to the Weaknesses section. I will adjust my rating according to the author's response.

**Ethical Concerns:**

["NO or VERY MINOR ethics concerns only"]

**Final Justification:**

The authors addressed my major concerns during rebuttal.

Given that there is still room for improvements in performance, latency, and computational costs compared to the baselines, I would suggest the rating of Borderline Accept after rebuttal.

**Limitations:**

The authors have briefly discussed the limitations.

**Quality:**

3

**Strengths And Weaknesses:**

### **Strengths**

1. RobIA presents a simple and efficient framework for real-time continual test-time adaptation (CTTA) in stereo depth estimation, addressing an important challenge for reliable real-world applications.
2. The author proposes a new benchmark from existing TTA sequences that feature frequent domain shifts. This benchmark more accurately reflects real-world conditions, where environmental changes occur regularly and previously encountered scenarios may reappear. It offers a more appropriate criterion for evaluating CTTA frameworks.
3. The paper is well-organized and easy to follow. Each component of the proposed framework is well-motivated and clearly explained.

### **Weaknesses**

1. **Lack of direct comparison with key baseline.** RobIA builds upon CoEx [1] stereo network, but the Modular ADaptation (MAD++) method from MADNet2 [2] is only evaluated on top of the original MADNet2 network. Comparing RobIA with MAD++ in this setting appears unfair. Furthermore, MAD++ with MADNet2 runs significantly faster than RobIA with the AdaptBN teacher (18ms vs. 204ms as reported in Table 3). A fairer comparison under similar runtime constraints (*e.g.*, by selecting additional modules in MAD++) would better clarify whether RobIA offers substantial improvements over the Modular ADaptation approach.
2. **Limited overall performance gain.** According to Table 1, 2 and 8 (in the supplementary material), the performance improvements with RobIA seem marginal compared to the AdaptBN approach with minimal trainable parameters (0.03M compared to 1.22M). Notably, as shown in Table 8, CoEx with AdaptBN even outperforms RobIA (w/ AdaptBN teacher) in KITTI RAW CTTA benchmark by a noticeable margin. This raises concerns about the effectiveness of RobIA.
3. **Minor presentation issues and typos.** I would suggest the author use `w/ Teacher` instead of `w/ AdaptBN` to reduce ambiguity with tunning batch normalization layers; In line 242, the loss should be $\mathcal{L}_{\text{teacher}}$ in the third equation; Bold and Underline fonts are missing from Tabel 2 onward.



[1] Bangunharcana, Antyanta, et al. "Correlate-and-excite: Real-time stereo matching via guided cost volume excitation." *2021 IEEE/RSJ International Conference on Intelligent Robots and Systems (IROS)*. IEEE, 2021.

[2] Poggi, Matteo, et al. "Continual adaptation for deep stereo." *IEEE Transactions on Pattern Analysis and Machine Intelligence* 44.9 (2021): 4713-4729.

---

> ### Author Rebuttal · Authors · 2025-07-31
>
> We appreciate the reviewer **4p4z** for recognizing the simplicity and efficiency of our CTTA framework, the realism of our benchmark design, and the clarity of our overall presentation.
>
>
> ---
>
> ## Weakness 1. Comparison with MAD++
>
> We thank the reviewer for highlighting the importance of a fair comparison with MAD++. To address this, we re-implemented our method, including AttEx-MoE and the AdaptBN teacher, on top of the MADNet2 [A] encoder, which is also used by MAD++ in Table 1, 2, and 8. Since MADNet2 lacks normalization layers, we inserted BatchNorm layers after each convolution to enable AdaptBN and maintain architectural consistency.
>
> **Table A** presents CTTA results using the MADNet2 backbone along with respective runtime, as requested.
>
> * On KITTI, which presents relatively mild domain shifts, (b) FULL++ achieves D1-all 1.13\% by updating all layers, while (d) our AttEx-MoE method achieves a better result (1.12\%) with fewer parameters. (e) RobIA, which combines AttEx-MoE with the AdaptBN teacher, maintains a strong KITTI score of 1.1\%.
>
> * On DrivingStereo, which exhibits stronger domain shifts, (b) FULL++ struggles with large domain shifts (6.13\%), and (c) MAD++ provides only minor improvement (5.86\%). In contrast, (d) our AttEx-MoE method further reduces the error to 5.53\%, and (e) RobIA achieves the best result at 4.83\%.
>
> These results show that our input-aware AttEx-MoE, combined with dual-source supervision, matches full tuning on stable domains (KITTI), and significantly outperforms both full and modular adaptation approaches under large distribution shifts (DrivingStereo).
>
> ### Table A
> |          |              |                                     | Mean          |          |            |          |         |
> |----------|--------------|-------------------------------------|---------------|----------|------------|----------|---------|
> |          |              |                                     | DrivingStereo |          | KITTI      |          | Runtime |
> | Backbone |              | Adapt.                              | D1-all (%)    | EPE (px) | D1-all (%) | EPE (px) | (ms)    |
> | MADNet2  |              | (a) Source                          | 10.44         | 1.7      | 3.92       | 1.09     | 11      |
> |          |              | (b) FULL++                          | 6.13          | 1.73     | 1.13       | 0.85     | 31      |
> |          |              | (c) MAD++                           | 5.86          | 1.41     | 1.24       | 0.88     | 18      |
> |          | (Ours)       | (d) AttEx-MoE                       | 5.53          | 1.20     | 1.12       | 0.86     | 23      |
> |          | (Ours-RobIA) | (e) AttEx-MoE w/ AdaptBN-Teacher    | 4.83          | 1.09     | 1.10       | 0.85     | 49      |
>
>
> ---
>
> ## Weakness 2. Limited overall performance gain
>
> ### **RobIA’s Gains Are Significant Under Realistic Domain Shifts.**
>
> We appreciate the concern regarding the seemingly marginal improvements of RobIA on KITTI. We agree that the performance gap may appear small. However, we would like to emphasize that this is primarily due to mild domain shifts and dense SGM pseudo-labels on KITTI, which make it easier to adapt even with lightweight methods like AdaptBN. In contrast, under more realistic and challenging conditions such as weather variation (DrivingStereo) and nighttime scenes (DSEC), RobIA demonstrates substantial improvements over AdaptBN.
>
> * In Tab. 1 and 2, (e) AdaptBN (0.03M trainable parameters) struggles under substantial domain shifts such as weather changes (DrivingStereo) and nighttime scenes (DSEC), yielding D1-all scores of 3.08\% and 4.54\%, respectively.
>
> * In contrast, (h) AttEx-MoE w/ AdaptBN-Teacher (1.22M) significantly improves performance to 2.77\% and 4.46\%.
>
>
> ### **KITTI Performance Reflects Domain Simplicity, Not Method Weakness.**
>
> AdaptBN performs competitively on KITTI (0.91\%, Tab.8), likely due to its (1) smaller domain gap (e.g., city $\rightarrow$ residential) and (2) higher SGM pseudo-label density (92\%) compared to DrivingStereo (72\%) and DSEC (45\%). These factors suggest that KITTI presents a more favorable adaptation setting, where even lightweight methods like AdaptBN can perform well.
>
> ### **Sequential Adaptation, KITTI $\rightarrow$ DrivingStereo, Further Highlights RobIA's Advantage.**
>
> To further support this point, we simulated a sequential CTTA scenario where the model first adapts to KITTI for 5 rounds, followed by adaptation to DrivingStereo for another 5 rounds, mimicking a realistic progression from a stable to a more challenging domain. Results are shown in **Table B**.
>
> * (b) and (c) show similar error rates with (a) on KITTI, but they achieve substantially lower error on DrivingStereo.
>
> * (a), despite strong performance of 1.11\% on KITTI, fails to adapt in the second phase (5.90\%).
>
> These results highlight the robustness of our AttEx-MoE design, which generalizes better under sequential, cross-domain conditions, mirroring realistic deployment settings.
>
>
> ### Table B
> |        |              |                                  | KITTI                   |          | DrivingStereo (after KITTI) |          | ALL        |          |
> |--------|--------------|----------------------------------|-------------------------|----------|-----------------------------|----------|------------|----------|
> |        |              |                                  | Round 1 $\rightarrow$ 5 |          | Round 6 $\rightarrow$ 10    |          |            |          |
> | Method |              | Adapt.                           | D1-all (%)              | EPE (px) | D1-all (%)                  | EPE (px) | D1-all (%) | EPE (px) |
> | CoEx   |              | (a) AdaptBN                      | 1.11                    | 0.84     | 5.9                         | 1.7      | 3.16       | 1.21     |
> |        | (Ours)       | (b) AttEx-MoE                    | 1.11                    | 0.85     | 4.22                        | 1.08     | 2.44       | 0.95     |
> |        | (Ours-RobIA) | (c) AttEx-MoE w/ AdaptBN-Teacher | 1.15                    | 0.84     | 2.66                        | 0.9      | 1.8        | 0.87     |
>
>
> ---
> ## Weakness 3. About Paper Formatting Concerns
>
> We appreciate your careful reading and pointing out the formatting issues. We will correct the loss term and fix the boldface formatting in Table 1 and the other tables. Additionally, we will rename the 'w/ AdaptBN' label to avoid ambiguity.
>
> ---
>
> ## Citations
> * [A] Tonioni, Alessio, et al. "Real-time self-adaptive deep stereo." Proceedings of the IEEE/CVF conference on computer vision and pattern recognition. 2019.

---

> > ### Comment · Reviewer_4p4z · 2025-08-01
> >
> > Thank the authors for providing additional comparison with MAD++ that addresses my major concern, and offering additional experiments that highlight RobIA's advantage in the sequential adaptation settings. I would suggest that the authors add these important results to the final manuscript.
> >
> > Given that there is still room for improvements in performance, latency, and computational costs compared to the baselines, I would suggest the rating of Borderline Accept after rebuttal.

---

> > > ### Author Response · Authors · 2025-08-02
> > >
> > > Thank you for your prompt feedback and for raising the score!
> > > We are truly happy we could address your concerns and are grateful for your thoughtful review. As per your suggestion, we will incorporate these new results into the final manuscript.

---

### Official Review · Reviewer_i8yG · 2025-06-30

**Clarity:** 3
**Significance:** 3
**Originality:** 2
**Rating:** 5
**Confidence:** 5

**Summary:**

This work studies the problem of tuning online a deep stereo network to unseen domains. The authors introduce two main novelties compared with previous works: the introduction of architecture modifications (mixture of experts) to make the model more robust to domain shifts and a new loss function using the prediction from a “teacher” model to speed up and stabilize convergence. The two modifications turned out to be effective across the different scenarios the author tested resulting in faster and more stable performance. The authors also introduce a new testing protocol by looping multiple times over the same video sequences. This setting should stress the ability of methods to adapt without catastrophic forgetting. The authors however do report performance also on the established way of testing test time trained stereo methods.

**Questions:**

* Are the timing in Tab. 3 taking into account the time spent generating pseudo labels using SGM or the AdaptBN teacher?

* [weakness b] Can you comment on weakness b?

**Ethical Concerns:**

["NO or VERY MINOR ethics concerns only"]

**Final Justification:**

Good work without major issue. The concern I raised in my initial review have been addressed. Overall I would suggest acceptance.

**Limitations:**

yes

**Paper Formatting Concerns:**

* Total Loss should have the third line being \lamda_teacher

* Tab. 1 bold need to be revised, all the other tables need bold

**Quality:**

3

**Strengths And Weaknesses:**

## Strengths

+ The test time training area is slightly under-explored in the depth estimation literature although it is one of the fields where it has proven to be most effective and practical. This work continues the existing exploration proposing both a new architecture which can be updated fast and a new loss that seems to be competitive and sometimes more effective than SOTA.

+  The use of a mixture of experts (MoE) in CNN architecture is another under-explored area. I found the proposal from the authors interesting and somehow novel, although I’m not entirely convinced in calling it a MoE since (per my understanding) there is no sparsity enforced and all “experts” are often active at the same time.

+ The work identifies one issue with existing SOTA around performance diverging for invalid regions and proposes an effective fix for it. The authors also designed  experiments to support their claim and solution (Fig. 2).

## Weaknesses

a. **Computational cost**: If my understanding is correct, the proposed solution relies on keeping two versions of the same model (one w/AdaptBN training and one with MoE training) and performing 2 fwd pass and 2 bwd pass for each frame being processed. This results in a 10x latency increase compared to pure inference (Tab. 3 row (j) vs row (d)) which is significantly more than the previous SOTA (row (b) and (c) ) in Tab. 3.

b. **Missing motivation/implementation details**: From the current version of the manuscript it is not clear to me why the MoE architecture proposed should help generalization. The parameters of the mixture will anyway be trained on the source domain so doing inference with the mixture should in theory suffer from domain shifts as much as the full trained model. Where the mixture might help is in regularizing test time training avoiding modifying the architecture too much, though the motivation presented in the work are different. Can the author clarify whether there is anything special in the way the mixture is trained that would help it generalize better.

C. [minor] **New evaluation protocol**: The authors propose a new evaluation protocol by looping over the same video sequence multiple times claiming that this scenario claiming to be the first method to look at test time adaptation for stereo under the "continuous" setting, while the previous methods operated under the assumption of a “single stationary target domain” (line 27-29). I would like to challenge this as the previous methods mentioned for SDE don’t have any assumption on the scenario considered and as such operate indeed on a continuous domain. Moreover a setting very similar to the one the authors considered is also reported in [1] Fig. 5 where the authors of [1] also test on looping over the same sequence. Challenging the motivation to develop a new test protocol would also imply that the methods should be compared using the old established protocol which the authors to their credits did in Tab. 9 Supplementary. In this table however the performance gains seem smaller with respect to the proposed new setting. This would benefit from some additional discussion in the document.

D. [minor] **Possible missing comparison**: [2] uses a combination of proxy labels loss + smoothing + image reconstruction to compensate for the sparsity of the proxy labels. This is exactly the problem the authors are trying to solve with using an AdaptBn teacher. While [2] tests this solution offline there is no reason why it cannot be run online. A comparison would be interesting as it would save to the authors the cost of running the expensive AdaptBN teacher.

## References

1. [Tonioni, Alessio, et al. "Real-time self-adaptive deep stereo." Proceedings of the IEEE/CVF conference on computer vision and pattern recognition. 2019.](https://openaccess.thecvf.com/content_CVPR_2019/papers/Tonioni_Real-Time_Self-Adaptive_Deep_Stereo_CVPR_2019_paper.pdf)
2. [Tonioni, Alessio, et al. "Unsupervised domain adaptation for depth prediction from images." IEEE transactions on pattern analysis and machine intelligence 42.10 (2019): 2396-2409.](https://arxiv.org/pdf/1909.03943)

---

> ### Author Rebuttal · Authors · 2025-07-31
>
> We appreciate the reviewer **i8yG** for highlighting the relevance of our test-time training setting, the novelty of our MoE-based design for CNNs, and our solution to performance degradation in invalid regions, along with the supporting experimental evidence.
>
> ---
> ## Weakness a. Computational Cost
>
> We respectfully point out that comparing (j) MoE w/ AdaptBN-Teacher to (d) CoEx Source in Tab. 3 is may not be entirely fair, as (d) represents a pure inference baseline without any adaptation, which falls outside the scope of test-time adaptation. While (j) incurs higher latency, it achieves the best overall performance (D1-all 2.77\%). More importantly, (i) our MoE-only variant strikes a favorable balance between performance and efficiency, achieving strong accuracy (D1-all 2.98\%) with moderate runtime (97.34ms). Practitioners may choose between (i) and (j) depending on the desired trade-off.
>
>
> ---
>
> ## Weakness b. How does the MoE architecture help generalization?
>
> We appreciate the reviewer’s question and the opportunity to clarify the motivation. AttEx-MoE improves generalization under domain shift by combining (1) geometry-aware, per-instance expert reweighting and (2) stable, soft channel modulation via Sigmoid gating, all within the PEFT setting. This design allows the model to adapt to instance-specific variations while preserving source knowledge through a frozen backbone.
>
> Specifically, the row-wise self-attention router extracts informative epipolar-consistent context, enabling input-aware reweighting of channel experts without modifying the backbone.
> More importantly, *Sigmoid-based gating* provides smooth and bounded scaling of expert activations on a per-instance basis. This is critical, as these gating values are directly multiplied with expert channels (See Eq. (2)).
> In contrast to ReLU, which may cause hard-zeroing and unbounded scaling, Sigmoid gating ensures bounded and stable modulation, preserving gradient flow and improving robustness under distribution shift.
> The effectiveness of Sigmoid gating function is empirically supported in Tab. 4, where it consistently outperforms ReLU across all router variants.
>
> ### Summary
> * AttEx-MoE enhances generalization via input-dependent expert routing and stable Sigmoid-based gating.
> * The frozen backbone preserves source knowledge, while the AttEx-MoE dynamically adapts to instance-specific features without architectural changes.
>
>
> ---
>
> ## Weakness c. New Evaluation protocol
>
> ### c-1. CTTA evaluation protocol
>
> We acknowledge that Fig. 5 in [A] evaluates model performance in a looping scenario. However, it operates within a *single domain* (Alley 2 from Sintel [B]) where the target distribution is fixed. While useful for analyzing long-term adaptation, it does not assess adaptation under domain shift or knowledge preserving across diverse environments.
>
> In contrast, our benchmark cycles through *multiple, distinct domains* (e.g., different weather conditions, nighttime scenes), introducing explicit distribution shifts. This allows us to evaluate continual adaptation challenges such as error accumulation and catastrophic forgetting, which fixed-domain setups cannot capture.
>
> We will clarify this distinction and cite [A] in the revised manuscript.
>
> ### c-2. TTA evaluation results
>
> As noted by the reviewer, (j) RobIA shows smaller gains under the TTA benchmark (Tab. 9-11) compared to (i) CoEx-FULL++. This can happen as TTA operates within a single domain using longer sequences, favoring full-tuning methods with higher capacity.
>
> This trend is also observed in MADNet 2, where (f) full tuning outperforms (g) modular update, particularly on KITTI ((f) 1.42\%, (g) 1.64\%) and DSEC ((f) 6.02\%, (g) 7.44\%), suggesting that high-capacity models benefit more from long-term adaptation in fixed-domain settings.
>
> Nevertheless, (j) RobIA achieves comparable performance to full-tuned baselines (See Tab. 9, 11).
> Interestingly, in DrivingStereo (Tab. 10), where intra-domain variability is higher (e.g., dusky, cloudy, rainy), (j) RobIA (5.02\%) outperforms (i) (5.2 \%), indicating stronger robustness under dynamic conditions.
>
>
> ---
>
> ## Weakness d. Comparison with Tonioni, Alessio, et al. 2019
>
> [C] introduces two additional self-supervised losses---a smoothness loss $L_{smooth}$ and a photometric reconstruction loss $L_{photo}$---to compensate for sparse proxy labels via auxiliary supervision. This shares a similar motivation as our AdaptBN teacher.
>
> To evaluate this strategy in our continual adaptation setting, we incorporated these loss terms into our framework as follows:
> $L$ = $L_{proxy}$ + $\lambda$ $\cdot$ $L_{smooth}$ + $\lambda$ $\cdot$ $L_{photo}$.
> In **Table A**, (a) and (e) corresponds to the baseline using only the proxy loss. (c) and (f) apply the above strategy with $\lambda=0.1$, but neither yields meaningful improvement over (a) and (d), respectively.
>
> In contrast, (b) and (e)---our AdaptBN-Teacher loss---provides more reliable pseudo-labels, thus lowering the error ((a) 3.04\% $\rightarrow$ (b) 2.93\%, (d) 2.98\% $\rightarrow$ (e) 2.77\%).
> This result confirms the effectiveness of our supervision strategy in continual adaptation.
>
> We will include this ablation in the revised manuscript and clarify that our teacher strategy outperforms simple photometric augmentation.
>
> ### Table A
> |                  |        |                                  | Mean   |      |
> |------------------|--------|----------------------------------|--------|------|
> | Backbone         | Source | Method                           | D1-all | EPE  |
> | CoEx             | Single | (a) $L_{proxy}$                  | 3.04   | 0.92 |
> |                  | Dual   | + (b) $L_{AdaptBN}$ (Ours)       | 2.93   | 0.91 |
> |                  |        | + (c) $L_{smooth}$ + $L_{photo}$ | 3.04   | 0.92 |
> | CoEx + AttEx-MoE | Single | (d) $L_{proxy}$                  | 2.98   | 0.97 |
> |                  | Dual   | + (e) $L_{AdaptBN}$ (Ours)       | 2.77   | 0.91 |
> |                  |        | + (f) $L_{smooth}$ + $L_{photo}$ | 2.98   | 0.96 |
>
>
> ---
> ## Question 1. Generation Time in Tab. 3
>
> In the timing in Tab. 3, we included pseudo-label generation through AdaptBN teacher. However, we excluded SGM label generation time for the same comparison with existing TTA methods [D], which also used SGM as a proxy label but do not account for its generation time in their runtime analysis.
>
> ---
> ## About Paper Formatting Concerns
>
> We appreciate your careful reading and pointing out the formatting issues. We will correct the loss term and fix the boldface formatting in Table 1 and the other tables.
>
> ---
>
> ## Citations
> * [A] Tonioni, Alessio, et al. "Real-time self-adaptive deep stereo." Proceedings of the IEEE/CVF conference on computer vision and pattern recognition. 2019.
> * [B] Butler, Daniel J., et al. "A naturalistic open source movie for optical flow evaluation." European conference on computer vision. Berlin, Heidelberg: Springer Berlin Heidelberg, 2012.
> * [C] Tonioni, Alessio, et al. "Unsupervised domain adaptation for depth prediction from images." IEEE transactions on pattern analysis and machine intelligence 42.10 (2019): 2396-2409.
> * [D] Poggi, Matteo, et al. "Continual adaptation for deep stereo." IEEE Transactions on Pattern Analysis and Machine Intelligence 44.9 (2021): 4713-4729.

---

> > ### Comment · Reviewer_i8yG · 2025-08-04
> > **Response to authors**
> >
> > Thanks for clarifying the questions I raised!

---

### Official Review · Reviewer_CJJ4 · 2025-06-30

**Clarity:** 2
**Significance:** 2
**Originality:** 2
**Rating:** 4
**Confidence:** 4

**Summary:**

The paper presents RobIA, a novel framework for Continual Test-time Adaptation (CTTA) in stereo depth estimation under dynamic domain shifts. The proposed method incorporates two key components: the Attend-and-Excite Mixture-of-Experts (AttEx-MoE) module for instance-aware adaptation and a Robust AdaptBN Teacher to enhance pseudo-label coverage. The authors show that RobIA outperforms existing methods in adapting to continually shifting domains, demonstrating strong performance in terms of both accuracy and computational efficiency. Overall, the approach is promising, and the experimental results are well-supported, though there are some areas for potential improvement.

**Questions:**

Please refer to weakness

**Ethical Concerns:**

["NO or VERY MINOR ethics concerns only"]

**Final Justification:**

Thanks for the response. I am satisfied with the authors' rebuttal, so I maintain my opinion of accepting the paper. The authors are advised to refine the related work section in the final version.

**Limitations:**

yes

**Quality:**

2

**Strengths And Weaknesses:**

Strengths:
Innovation and Novelty: The introduction of AttEx-MoE and the dual-source pseudo-supervision strategy is a contribution to the field of continual adaptation for stereo depth estimation. The use of instance-aware, epipolar geometry-guided adaptation offers a unique approach to overcoming domain shifts in real-world settings.

Experimental Evaluation: The authors conduct extensive experiments across several challenging benchmarks (KITTI RAW, DrivingStereo, and DSEC), showing that RobIA performs robustly in dynamic, real-world conditions. The results demonstrate superior adaptation performance when compared to existing methods like MADNet2 and CoEx.

Efficient Use of Resources: RobIA achieves high performance while maintaining computational efficiency, which is essential for real-time applications such as autonomous driving. The proposed architecture leverages parameter-efficient modules, which helps in maintaining a balance between accuracy and resource usage.


Weakness:
1.In the KITTI15 leaderboard, among the top 15 models, 5 models with attached paper links include MonSter[1], DEFOM-Stereo[2], IGEV++ (DepthAny.)[3], StereoBase[4], and TC-Stereo[5]. MonSter[1], DEFOM-Stereo[2], and IGEV++ (DepthAny.)[3] utilize monocular Foundation models. IGEV[6] is a recently widely experimented baseline for stereo tasks, so the review recommends verifying the effectiveness of the method proposed in the paper by choosing one of IGEV or StereoBase. To the best knowledge of review, LightStereo[7] is the SOTA 2D-based model. The reviewer believes that the authors need to demonstrate the effectiveness of their method using at least one of the above methods.

[1] Cheng, Junda, et al. "MonSter: Marry Monodepth to Stereo Unleashes Power." (CVPR2025).

[2] Jiang, Hualie, et al. "DEFOM-Stereo: Depth Foundation Model Based Stereo Matching."  (CVPR2025).

[3] G. Xu, X. Wang, Z. Zhang, J. Cheng, C. Liao and X. Yang: IGEV++: Iterative Multi-range Geometry Encoding Volumes for Stereo Matching. IEEE TPAMI 2025.

[4] X. Guo, J. Lu, C. Zhang, Y. Wang, Y. Duan, T. Yang, Z. Zhu and L. Chen: OpenStereo: A Comprehensive Benchmark for Stereo Matching and Strong Baseline. arXiv preprint arXiv:2312.00343 2023.

[5] J. Zeng, C. Yao, Y. Wu and Y. Jia: Temporally Consistent Stereo Matching. （ECCV2024).

[6]  Xu, Gangwei, et al. "Iterative geometry encoding volume for stereo matching." (CVPR2023).

[7] "Lightstereo: Channel boost is all you need for efficient 2d cost aggregation." (ICRA 2025).

2. In Tables 2, 3 and 4, the best results need to be bolded to facilitate readers' reading.

---

> ### Author Rebuttal · Authors · 2025-07-31
>
> We thank the reviewer **CJJ4** for their thoughtful feedback. We are encouraged that they found our idea innovative and novel, supported by extensive experiments and efficient use of limited resources.
>
> ---
>
> ## Weakness 1. Broader Backbone Evaluation
>
> We appreciate the suggestion regarding evaluation using additional competitive architectures. In response, we extended our experiments to two widely recognized and publicly available backbones:
>
> * **IGEV-Stereo** [A]: a widely adopted iterative refinement model
> * **LightStereo** [B]: a lightweight model with 2D cost aggregation
>
>
> ### 1-1. Evaluation on IGEV-Stereo
>
> As shown in **Table A**, our method significantly improves performance. These results confirm that our plug-and-play modules effectively enhance generalization even on top-performing backbones.
>
> * (a) IGEV-Source suffers from domain shift (7.26\% D1-all on DrivingStereo).
> * (b) AdaptBN tuning only BN parameters and (c) FULL++ still struggle for adaptation (3.87\%, 3.56\%, respectively).
> * (d) Our AttEx-MoE reduces DrivingStereo error to 3.15\%.
> * (e) The proposed method, RobIA (MoE w/ AdaptBN-Teacher), achieves the best performance: 2.76\% on DrivingStereo.
>
>
> #### Table A
> |             |              |                                     | Mean          |          |
> |-------------|--------------|-------------------------------------|---------------|----------|
> |             |              |                                     | DrivingStereo |          |
> | Backbone    |              | Adapt.                              | D1-all (%)    | EPE (px) |
> | IGEV-Stereo |              | (a) Source                          | 7.26          | 1.32     |
> |             |              | (b) AdaptBN                         | 3.87          | 0.98     |
> |             |              | (c) FULL++                          | 3.56          | 0.96     |
> |             | (Ours)       | (d) AttEx-MoE                       | 3.15          | 0.92     |
> |             | (Ours-RobIA) | (e) AttEx-MoE w/ AdaptBN-Teacher    | 2.76          | 0.87     |
>
>
> ### 1-2. Evaluation on LightStereo
>
> RobIA also generalizes well to LightStereo, a compact real-time model. The results are shown in **Table B**.
>
> * (a) LightStereo-Source shows severe degradation under domain shift (18.59\%).
> * (b) AdaptBN improves accuracy (5.95\%) but remains limited on dynamic scenes.
> * (d) Our AdaptBN-Teacher improves CoEx FULL++ to 5.18\%, reinforcing the strength of our dense pseudo-labeling strategy.
> * (e) AttEx-MoE alone reduces DrivingStereo error to 5.76\%.
> * (f) The proposed method, RobIA, achieves the best results: 4.95\% on DrivingStereo.
>
>
> #### Table B
> |             |              |                                     | Mean          |          |
> |-------------|--------------|-------------------------------------|---------------|----------|
> |             |              |                                     | DrivingStereo |          |
> | Backbone    |              | Adapt.                              | D1-all (%)    | EPE (px) |
> | LightStereo |              | (a) Source                          | 18.59         | 3.11     |
> |             |              | (b) AdaptBN                         | 5.95          | 1.74     |
> |             |              | (c) FULL++                          | 5.91          | 1.72     |
> |             |              | (d) FULL++ w/ AdaptBN-Teacher       | 5.18          | 1.13     |
> |             | (Ours)       | (e) AttEx-MoE                       | 5.76          | 1.28     |
> |             | (Ours-RobIA) | (f) AttEx-MoE w/ AdaptBN-Teacher    | 4.95          | 1.11     |
>
> ---
> ## Weakness 2. About Paper Formatting Concerns
>
> We appreciate your careful reading and pointing out the formatting issues. We will correct the loss term and fix the boldface formatting in Table 1 and the other tables.
>
> ## Citations
> * [A] Xu, Gangwei, et al. "Iterative geometry encoding volume for stereo matching." Proceedings of the IEEE/CVF conference on computer vision and pattern recognition. 2023.
> * [B] Guo, Xianda, et al. "LightStereo: Channel Boost Is All You Need for Efficient 2D Cost Aggregation." arXiv preprint arXiv:2406.19833 (2024).

---

> > ### Comment · Reviewer_CJJ4 · 2025-08-02
> >
> > Thanks for the response. I am satisfied with the authors' rebuttal, so I maintain my opinion of accepting the paper. The authors are advised to refine the related work section in the final version.

---

> > > ### Author Response · Authors · 2025-08-03
> > >
> > > Thank you for your prompt feedback and for accepting our paper! We are truly happy we could address your concerns and are grateful for your thoughtful review. As per your suggestion, we will incorporate these new results into the final manuscript.
> > > Additionally, we will refine our related work section as advised.

---

### Note · Authors · 2025-08-16

We sincerely thank the reviewers for their constructive feedback. We are encouraged by the recognition of RobIA’s novelty, practicality, and clarity, as well as the unanimous recommendation for acceptance.

## Reviewers noted the following key strengths:

- The innovation of AttEx-MoE and the dual-source pseudo-supervision strategy was recognized as a novel contribution to the under-explored area of continual test-time adaptation (CTTA) in stereo depth estimation (SDE).
- The proposed architecture was endorsed for its robustness and effectiveness in overcoming domain shifts in real-world scenarios.
- Importantly, the work identifies a key limitation of existing SOTA methods, i.e. performance degradation in invalid regions, and addresses it through a principled solution, supported by extensive experiments.
- The newly proposed CTTA benchmark, introducing frequent domain shifts across diverse datasets (KITTI RAW, DrivingStereo, DSEC), was acknowledged as a valuable step toward more realistic evaluation. Notably, our method showed strong performance on it.

## Reviewer concerns and our responses

We also appreciate their valuable comments, and are pleased to report that **these comments were successfully addressed**. The comments included the need for broader evaluation on SOTA models (**CJJ4**), clarification of the AttEx-MoE motivation (**i8yG**), fairer comparisons with key baselines and justification for limited performance gain in KITTI RAW (**4p4z**), and applicability to architectures without batch normalization (**VsX9**).

In response:

- We expanded our experimental coverage by incorporating additional SOTA SDE models, including IGEV-Stereo and LightStereo (**CJJ4**), and conducted a fair comparison against MADNet2, the key baseline (**4p4z**).
- We clarified the motivation behind AttEx-MoE as an instance-aware MoE designed for PEFT in CNN-based stereo architectures (**i8yG**).
- We further analyzed RobIA’s performance improvement under realistic sequential domain shifts (including KITTI RAW), reinforcing its robustness in dynamic adaptation scenarios (**4p4z**).
- We validated the generality of the AdaptBN-Teacher through its successful application to MADNet2, highlighting its robustness in the absence of normalization (**VsX9**).

We believe that RobIA provides a robust and extensible foundation for continual adaptation in real-world dense prediction tasks, and we hope it will serve as a valuable cornerstone for future research in this area.

---

### Decision · Program_Chairs · 2025-09-17

**Decision:**

Accept (poster)

**Comment:**

This paper proposes a continual Test-time Adaptation in stereo depth estimation, with two key components, two key components: the Attend-and-Excite Mixture-of-Experts (AttEx-MoE) module for instance-aware adaptation and a Robust AdaptBN Teacher to enhance pseudo-labels. After the rebuttal, the reviewers were generally positive to the paper’s contribution. Their raised major concerns regarding unclarified motivation/complexity analysis/fairness in comparison, generalization to other methods and architectures, required additional comparisons, were satisfactory addressed in the rebuttal, as acknowledged by all the reviewers. After reading the paper, the comments and the rebuttal, the AC agreed with the reviewers’ consensus that the paper could be accepted. The AC reminds the authors to incorporate the committed revisions in the rebuttal to the final version.